# Jumping Ahead: Improving Reconstruction Fidelity with JumpReLU Sparse Autoencoders

## Abstract

Sparse autoencoders (SAEs) are a promising unsupervised approach for identifying causally relevant and interpretable linear features in a language model's (LM) activations. To be useful for downstream tasks, SAEs need to decompose LM activations faithfully; yet to be interpretable the decomposition must be sparse – two objectives that are in tension. In this paper, we introduce JumpReLU SAEs, which achieve state-of-the-art reconstruction fidelity at a given sparsity level on Gemma 2 9B activations, compared to other recent advances such as Gated and TopK SAEs. We also show that this improvement does not come at the cost of interpretability through manual and automated interpretability studies. JumpReLU SAEs are a simple modification of vanilla (ReLU) SAEs – where we replace the ReLU with a discontinuous JumpReLU activation function – and are similarly efficient to train and run. By utilizing straight-through-estimators (STEs) in a principled manner, we show how it is possible to train JumpReLU SAEs effectively despite the discontinuous JumpReLU function introduced in the SAE's forward pass. Similarly, we use STEs to directly train L0 to be sparse, instead of training on proxies such as L1, avoiding problems like shrinkage.

## 1 Introduction

Sparse autoencoders (SAEs) allow us to find causally relevant and seemingly interpretable directions in the activation space of a language model (Bricken et al., 2023; Cunningham et al., 2023; Templeton et al., 2024). There is interest within the field of mechanistic interpretability in using sparse decompositions produced by SAEs for tasks such as circuit analysis (Marks et al., 2024) and model steering (Conmy & Nanda, 2024).

SAEs work by finding approximate, sparse, linear decompositions of language model (LM) activations in terms of a large dictionary of basic "feature" directions. Two key objectives for a good decomposition (Bricken et al., 2023) are that it is sparse – i.e. that only a few elements of the dictionary are needed to reconstruct any given activation – and that it is faithful – i.e. the approximation error between the original activation and recombining its SAE decomposition is "small" in some suitable sense. These two objectives are naturally in tension: for any given SAE training method and fixed dictionary size, increasing sparsity typically results in reduced reconstruction fidelity.

One strand of recent research in training SAEs on LM activations (Rajamanoharan et al., 2024; Gao et al., 2024; Taggart, 2024) has been on finding improved SAE architectures and training methods that push out the Pareto frontier balancing these two objectives, while preserving other less quantifiable measures of SAE quality such as the interpretability or functional relevance of dictionary directions. A common thread connecting these recent improvements is the introduction of a thresholding or gating operation to determine which SAE features to use in the decomposition. The motivation for introducing such a thresholding operation is illustrated and explained in Fig. 1.

In this paper, we introduce **JumpReLU SAEs** – a small modification of the original, ReLU-based SAE architecture (Ng, 2011) where the SAE encoder's ReLU activation function is replaced by a JumpReLU activation function (Erichson et al., 2019), which zeroes out pre-activations below a positive threshold (see Fig. 1). Moreover, we train JumpReLU SAEs using a loss function that is simply the weighted sum of a L2 reconstruction error term and a L0 sparsity penalty, eschewing

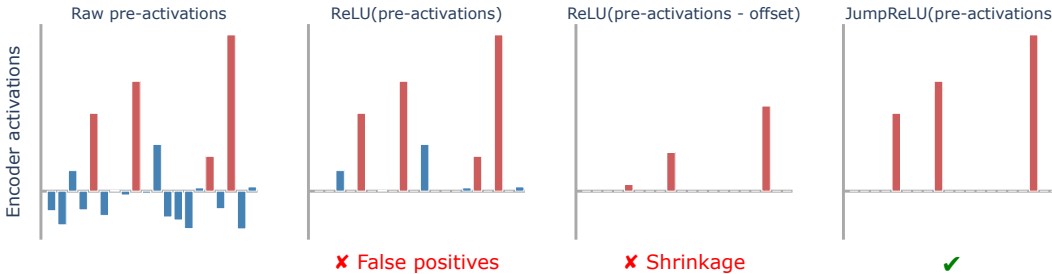

Figure 1: A toy model illustrating why JumpReLU (or similar activation functions, such as TopK) are an improvement over ReLU for training sparse yet faithful SAEs. Consider a direction in which the encoder pre-activation is high when the corresponding feature is **active** and low, but not always negative, when the feature is **inactive** (far-left). Applying a ReLU activation function fails to remove all false positives (centre-left), harming sparsity. We can get rid of false positives while maintaining the ReLU, e.g. by decreasing the encoder bias (centre-right), but this leads to feature magnitudes being systematically underestimated, harming fidelity. The JumpReLU activation function (far-right) provides an independent threshold below which pre-activations are screened out, minimizing false positives, while leaving pre-activations above the threshold unaffected, improving fidelity.

easier-to-train proxies to L0, such as L1, and without needing auxiliary tasks to train the threshold (which Gated SAEs require).

Our key insight is to notice that although such a loss function is piecewise-constant with respect to the threshold – and therefore provides zero gradient to train this parameter – the derivative of the *expected loss* can be analytically derived, and is generally non-zero, albeit it is expressed in terms of probability densities of the feature activation distribution that need to be estimated. We show how to use straight-through-estimators (STEs; Bengio et al. (2013)) to estimate the gradient of the expected loss in an efficient manner, thus allowing JumpReLU SAEs to be trained using standard gradient-based methods.

We evaluate JumpReLU, Gated and TopK (Gao et al., 2024) SAEs on Gemma 2 9B (Gemma Team, 2024) residual stream, MLP output and attention output activations at several layers (Fig. 2). At any given level of sparsity, we find JumpReLU SAEs consistently provide more faithful reconstructions than Gated SAEs. JumpReLU SAEs also provide reconstructions that are at least as good as, and often slightly better than, TopK SAEs. Similar to simple ReLU SAEs, JumpReLU SAEs only require a single forward and backward pass during a training step and have an elementwise activation function (unlike TopK, which requires a partial sort), making them more efficient to train than either Gated or TopK SAEs.

Compared to Gated SAEs, we find both TopK and JumpReLU tend to have more features that activate very frequently – i.e. on more than 10% of tokens (Fig. 4). Consistent with prior work evaluating TopK SAEs (Cunningham & Conerly, 2024) we find these high frequency JumpReLU features tend to be less interpretable, although interpretability does improve as SAE sparsity increases. Furthermore, only a small proportion of SAE features have very high frequencies: fewer than 0.06% in a 131k-width SAE. We also present the results of manual and automated interpretability studies indicating that randomly chosen JumpReLU, TopK and Gated SAE features are similarly interpretable.

## 2 PRELIMINARIES

**SAE architectures** SAEs sparsely decompose language model activations $\mathbf{x} \in \mathbb{R}^n$ as a linear combination of a *dictionary* of $M \gg n$ *learned feature* directions and then reconstruct the original activations using a pair of encoder and decoder functions $(\mathbf{f}, \hat{\mathbf{x}})$ defined by:

$$\mathbf{f}(\mathbf{x}) := \sigma\left(\mathbf{W}_{\text{enc}}\mathbf{x} + \mathbf{b}_{\text{enc}}\right); \quad \hat{\mathbf{x}}(\mathbf{f}) := \mathbf{W}_{\text{dec}}\mathbf{f} + \mathbf{b}_{\text{dec}}. \quad (1)$$

In these expressions, $\mathbf{f}(\mathbf{x}) \in \mathbb{R}^M$ is a sparse, non-negative vector of feature magnitudes present in the input activation $\mathbf{x}$, whereas $\hat{\mathbf{x}}(\mathbf{f}) \in \mathbb{R}^n$ is a reconstruction of the original activation from a feature representation $\mathbf{f} \in \mathbb{R}^M$. The columns of $\mathbf{W}_{\text{dec}}$, which we denote by $\mathbf{d}_i$ for $i = 1 \ldots M$,

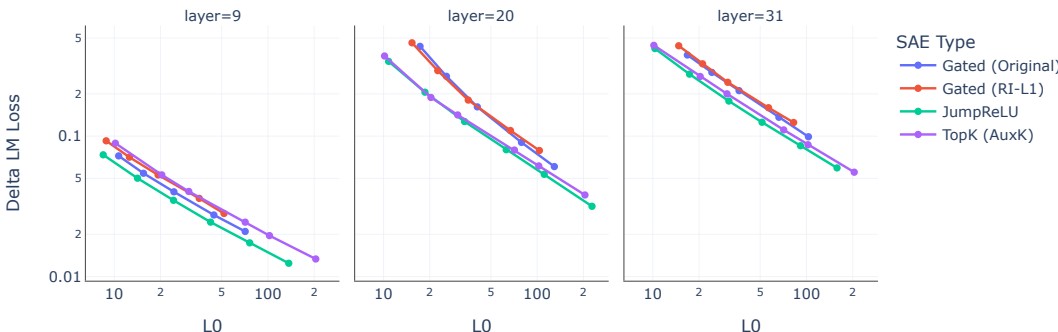

Figure 2: JumpReLU SAEs offer reconstruction fidelity, as measured by delta LM loss, that equals or exceeds Gated and TopK SAEs at a fixed level of sparsity, as measured by L0. These results are for SAEs trained on the residual stream after layers 9, 20 and 31 of Gemma 2 9B. See Fig. 14 and Fig. 15 for analogous plots for SAEs trained on MLP and attention output activations at these layers. Delta LM loss is the increase in cross-entropy loss incurred when the SAE is spliced into the LM, whereas L0 is the average number of SAE features that are active. Log scales are used on both axes.

represent the dictionary of directions into which the SAE decomposes $\mathbf{x}$. We also use $\boldsymbol{\pi}(\mathbf{x})$ in this text to denote the encoder's pre-activations, $\boldsymbol{\pi}(\mathbf{x}) := \mathbf{W}_{\text{enc}}\mathbf{x} + \mathbf{b}_{\text{enc}}$.

**Activation functions**  The activation function $\sigma$ varies between architectures: Bricken et al. (2023) and Templeton et al. (2024) use the ReLU activation function, whereas TopK SAEs (Gao et al., 2024) use a TopK activation function (which zeroes out all but the top $K$ pre-activations). Gated SAEs (Rajamanoharan et al., 2024) in their general form do not fit the specification of Eq. (1); however with weight sharing between the two encoder kernels, they can be shown (Rajamanoharan et al., 2024, Appendix E) to be equivalent to using a JumpReLU activation function, defined as

$$\text{JumpReLU}_{\theta}(z) := z\,H(z - \theta) \tag{2}$$

where $H$ is the Heaviside step function[1] when $\theta > 0$ is the JumpReLU's threshold, below which pre-activations are set to zero, as shown on the lefthand side of Fig. 3.

**Loss functions**  Language model SAEs are trained to reconstruct samples from a large dataset of language model activations $\mathbf{x} \sim \mathcal{D}$ typically using a loss function of the form

$$\mathcal{L}(\mathbf{x}) := \underbrace{\|\mathbf{x} - \hat{\mathbf{x}}(\mathbf{f}(\mathbf{x}))\|_2^2}_{\mathcal{L}_{\text{reconstruct}}} + \underbrace{\lambda\,S(\mathbf{f}(\mathbf{x}))}_{\mathcal{L}_{\text{sparsity}}} + \mathcal{L}_{\text{aux}}, \tag{3}$$

where $S$ is a function of the feature coefficients that penalizes non-sparse decompositions and the *sparsity coefficient* $\lambda$ sets the trade-off between sparsity and reconstruction fidelity. Optionally, auxiliary terms in the loss function, $\mathcal{L}_{\text{aux}}$ may be included for a variety of reasons, e.g. to help train parameters that would otherwise not receive suitable gradients (used for Gated SAEs) or to resurrect unproductive ("dead") feature directions (used for TopK). Note that TopK SAEs are trained without a sparsity penalty, since the TopK activation function directly enforces sparsity.

**Sparsity penalties**  Both the ReLU SAEs of Bricken et al. (2023) and Gated SAEs use the L1-norm $S(\mathbf{f}) := \|\mathbf{f}\|_1$ as a sparsity penalty. While this has the advantage of providing a useful gradient for training (unlike the L0-norm), it has the disadvantage of penalizing feature magnitudes in addition to sparsity, which harms reconstruction fidelity (Rajamanoharan et al., 2024; Wright & Sharkey, 2024).

The L1 penalty also fails to be invariant under reparameterizations of a SAE; by scaling down encoder parameters and scaling up decoder parameters accordingly, it is possible to arbitrarily shrink feature magnitudes, and thus the L1 penalty, without changing either the number of active features or the SAE's output reconstructions. As a result, it is necessary to impose a further constraint

---

[1] $H(z)$ is one when $z > 0$ and zero when $z < 0$. Its value when $z = 0$ is a matter of convention – unimportant when $H$ appears within integrals or integral estimators, as is the case in this paper.

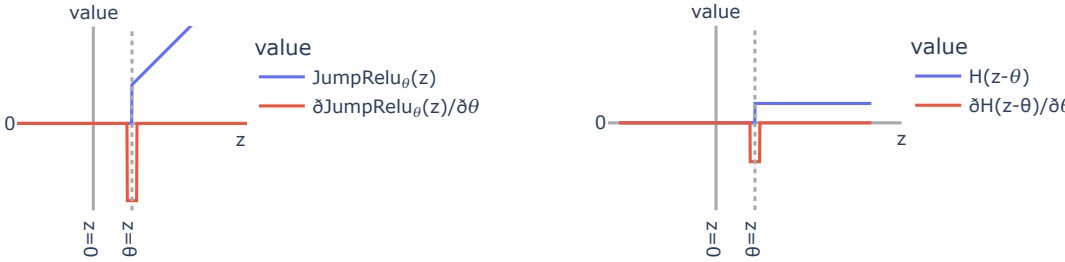

Figure 3: The JumpReLU activation function (left) and the Heaviside step function (right) used to calculate the sparsity penalty are piecewise constant with respect to the JumpReLU threshold. There-fore, in order to be able to train a JumpReLU SAE, we define the pseudo-derivatives illustrated in these plots and defined in Eq. (8) and Eq. (9), which approximate the Dirac delta functions present in the actual (weak) derivatives of the JumpReLU and Heaviside functions. These pseudo-derivatives provide a gradient signal to the threshold whenever pre-activations are within a small window of width $\varepsilon$ around the threshold. Note these plots show the profile of these pseudo-derivatives in the $z$, not $\theta$ direction, as $z$ is the stochastic input that is averaged over when computing the mean gradient.

on SAE parameters during training to enforce sparsity: typically this is achieved by constraining columns of the decoder weight matrix $\mathbf{d}_i$ to have unit norm (Bricken et al., 2023). Conerly et al. (2024) introduce a modification of the L1 penalty, where feature coefficients are weighted by the norms of the corresponding dictionary directions, i.e. $S_{\text{RI-L1}}(\mathbf{f}) := \sum_{i=1}^{M} f_i \|\mathbf{d}_i\|_2$. We call this the *reparameterization-invariant L1* (RI-L1) sparsity penalty, since this penalty is invariant to SAE reparameterization, making it unnecessary to impose constraints on $\|\mathbf{d}_i\|_2$.

**Kernel density estimation**  Kernel density estimation (KDE; Parzen (1962); Wasserman (2010)) is a technique for empirically estimating probability densities from a finite sample of observations. Given $N$ samples $x_{1\ldots N}$ of a random variable $X$, one can form a kernel density estimate of the probability density $p_X(x)$ using an estimator of the form $\hat{p}_X(x) := \frac{1}{N\varepsilon} \sum_{\alpha=1}^{N} K\left(\frac{x-x_\alpha}{\varepsilon}\right)$, where $K$ is a non-negative function that satisfies the properties of a centred, positive-variance probability density function and $\varepsilon$ is the kernel *bandwidth* parameter.[2] In this paper we will actually be inter-ested in estimating quantities like $v(y) = \mathbb{E}[f(X,Y)|Y = y]p_Y(y)$ for jointly distributed random variables $X$ and $Y$ and arbitrary (but well-behaved) functions $f$. Following a similar derivation as in Wasserman (2010, Chapter 20), it is straightforward to generalize KDE to estimate $v(y)$ using the estimator

$$\hat{v}(y) := \frac{1}{N\varepsilon} \sum_{\alpha=1}^{N} f(x_\alpha, y_\alpha) K\left(\frac{y - y_\alpha}{\varepsilon}\right). \tag{4}$$

## 3   JUMPRELU SAEs

A JumpReLU SAE is a SAE of the standard form Eq. (1) with a JumpReLU activation function:

$$\mathbf{f}(\mathbf{x}) := \text{JumpReLU}_{\boldsymbol{\theta}}\left(\mathbf{W}_{\text{enc}}\mathbf{x} + \mathbf{b}_{\text{enc}}\right). \tag{5}$$

Compared to a ReLU SAE, it has an extra positive vector-valued parameter $\boldsymbol{\theta} \in \mathbb{R}_+^M$ that specifies, for each feature $i$, the threshold that encoder pre-activations need to exceed in order for the feature to be deemed active.

Similar to the gating mechanism in Gated SAEs and the TopK activation function in TopK SAEs, the threshold $\boldsymbol{\theta}$ gives JumpReLU SAEs the means to separate out deciding which features are active from estimating active features' magnitudes, as illustrated in Fig. 1.

We train JumpReLU SAEs using the loss function

$$\mathcal{L}(\mathbf{x}) := \underbrace{\|\mathbf{x} - \hat{\mathbf{x}}(\mathbf{f}(\mathbf{x}))\|_2^2}_{\mathcal{L}_{\text{reconstruct}}} + \underbrace{\lambda \|\mathbf{f}(\mathbf{x})\|_0}_{\mathcal{L}_{\text{sparsity}}}. \tag{6}$$

---

[2]I.e. $K(x) \geq 0$, $\int_{-\infty}^{\infty} K(x)\mathrm{d}x = 1$, $\int_{-\infty}^{\infty} x\,K(x)\mathrm{d}x = 0$ and $\int_{-\infty}^{\infty} x^2 K(x)\mathrm{d}x > 0$.

This is a loss function of the standard form Eq. (3) where crucially we are using a L0 sparsity penalty to avoid the limitations of training with a L1 sparsity penalty (Wright & Sharkey, 2024; Rajamanoharan et al., 2024). Note that we can also express the L0 sparsity penalty in terms of a Heaviside step function on the encoder's pre-activations $\boldsymbol{\pi}(\mathbf{x})$:

$$\mathcal{L}_{\text{sparsity}} := \lambda \left\| \mathbf{f}(\mathbf{x}) \right\|_0 = \lambda \sum_{i=1}^{M} H(\pi_i(\mathbf{x}) - \theta_i). \tag{7}$$

The difficulty with training using this loss function is that it provides no gradient signal for training the threshold: $\boldsymbol{\theta}$ appears only within the arguments of Heaviside step functions in both $\mathcal{L}_{\text{reconstruct}}$ and $\mathcal{L}_{\text{sparsity}}$ (as expressed in Eq. (7)).[3] Our solution is to use straight-through-estimators (STEs; Bengio et al. (2013)), as illustrated in Fig. 3. Specifically, we define the following pseudo-derivative for JumpReLU$_\theta(z)$:[4]

$$\frac{\eth}{\eth\theta} \text{JumpReLU}_\theta(z) := -\frac{\theta}{\varepsilon} K\left( \frac{z - \theta}{\varepsilon} \right) \tag{8}$$

and the following pseudo-derivative for the Heaviside step function appearing in the L0 penalty:

$$\frac{\eth}{\eth\theta} H(z - \theta) := -\frac{1}{\varepsilon} K\left( \frac{z - \theta}{\varepsilon} \right). \tag{9}$$

In these expressions, $K$ can be any valid kernel function (see Section 2) – i.e. it needs to satisfy the properties of a centered, finite-variance probability density function. In our experiments, we use the rectangle function, $\text{rect}(z) := H\left(z + \frac{1}{2}\right) - H\left(z - \frac{1}{2}\right)$ as our kernel; however similar results can be obtained with other common kernels, such as the triangular, Gaussian or Epanechnikov kernel (see Appendix H.3). As we show in Section 4, the hyperparameter $\varepsilon$ plays the role of a KDE bandwidth, and needs to be selected accordingly: too low and gradient estimates become too noisy, too high and estimates become too biased.[5]

Having defined these pseudo-derivatives, we train JumpReLU SAEs as we would any differentiable model, by computing the gradient of the loss function in Eq. (6) over batches of data (remembering to apply these pseudo-derivatives in the backward pass), and sending the batch-wise mean of these gradients to the optimizer in order to compute parameter updates.

In Appendix K we provide pseudocode for the JumpReLU SAE's forward pass, loss function and for implementing the straight-through-estimators defined in Eq. (8) and Eq. (9) in an autograd framework like Jax (Bradbury et al., 2018) or PyTorch (Paszke et al., 2019).

## 4 HOW STES ENABLE TRAINING THROUGH THE JUMP

Why does this work? The key is to notice that during SGD, we actually want to estimate the gradient of the *expected* loss, $\mathbb{E}_{\mathbf{x}} [\mathcal{L}_{\boldsymbol{\theta}}(\mathbf{x})]$, in order to calculate parameter updates.[6] Although the loss itself is piecewise constant with respect to the threshold parameters – and therefore has zero gradient – the expected loss is not.

As shown in Appendix B, we can differentiate expected loss with respect to $\boldsymbol{\theta}$ analytically to obtain

$$\frac{\partial \mathbb{E}_{\mathbf{x}} [\mathcal{L}_{\boldsymbol{\theta}}(\mathbf{x})]}{\partial \theta_i} = \left( \mathbb{E}_{\mathbf{x}} [I_i(\mathbf{x}) | \pi_i(\mathbf{x}) = \theta_i] - \lambda \right) p_i(\theta_i), \tag{10}$$

---

[3]The L0 sparsity penalty also provides no gradient signal for the remaining SAE parameters, but this is not necessarily a problem. It just means that the remaining SAE parameters are encouraged purely to reconstruct input activations faithfully, not worrying about sparsity, while sparsity is taken care of by the threshold parameter $\boldsymbol{\theta}$. This is analogous to TopK SAEs, where similarly the main SAE parameters are trained solely to reconstruct faithfully, while sparsity is enforced by the TopK activation function.

[4]We use the notation $\eth/\eth\theta$ to denote pseudo-derivatives, to avoid conflating them with actual partial derivatives for these functions. In principle, we could also define pseudo-derivatives of with respect to $z$, i.e. to provide gradient signals from the JumpReLU threshold and L0 sparsity penalty to the encoder parameters in addition to the gradients to the threshold provided by Eqs. (8) and (9). However, as explored in Appendix H.1, we find this leads to SAEs plagued by dead features and with poor reconstruction fidelity.

[5]For the experiments in this paper, we swept this parameter and found $\varepsilon = 0.001$ (assuming a dataset normalized such that $\mathbb{E}_{\mathbf{x}}[\mathbf{x}^2] = 1$) works well across different models, layers and sites. However, we suspect there are more principled ways to determine this parameter.

[6]In this section, we write the JumpReLU loss as $\mathcal{L}_{\boldsymbol{\theta}}(\mathbf{x})$ to show explicitly its dependence on $\boldsymbol{\theta}$.

where $p_i$ is the probability density function for the distribution of feature pre-activations $\pi_i(\mathbf{x})$ and

$$I_i(\mathbf{x}) := 2\theta_i \mathbf{d}_i \cdot (\mathbf{x} - \hat{\mathbf{x}}(\mathbf{f}(\mathbf{x}))), \tag{11}$$

recalling that $\mathbf{d}_i$ is the column of $\mathbf{W}_{\text{dec}}$ corresponding to feature $i$.[7]

In order to train JumpReLU SAEs, we need to estimate the gradient as expressed in Eq. (10) from batches of input activations, $\mathbf{x}_1, \mathbf{x}_2, \ldots, \mathbf{x}_N$. To do this, we can use a generalized KDE estimator of the form Eq. (4). This gives us the following estimator of the expected loss's gradient with respect to $\boldsymbol{\theta}$:

$$\frac{1}{N\varepsilon} \sum_{\alpha=1}^{N} \{I_i(\mathbf{x}_\alpha) - \lambda\} K\left(\frac{\pi_i(\mathbf{x}_\alpha) - \theta_i}{\varepsilon}\right). \tag{12}$$

As we show in Appendix C, when we instruct autograd to use the pseudo-derivatives defined in Eqs. (8) and (9) in the backward pass, this is precisely the batch-wise mean gradient that gets calculated – and used by the optimizer to update $\boldsymbol{\theta}$ – in the training loop.

In other words, training with straight-through-estimators as described in Section 3 is equivalent to estimating the true gradient of the expected loss, as given in Eq. (10), using the kernel density estimator defined in Eq. (12).

## 5 EVALUATION

In this section, we compare JumpReLU SAEs to Gated and TopK SAEs across a range of metrics.[8] To make these comparisons, we trained multiple 131k-width SAEs (with a range of sparsity levels) of each type (JumpReLU, Gated and TopK) on activations from Gemma 2 9B (base). Specifically, we trained SAEs on residual stream, attention output and MLP output sites after layers 9, 20 and 31 of the model (zero-indexed), i.e. approximately $1/4$, $1/2$ and $3/4$-way through this 42-layer model.

We trained Gated SAEs using two different loss functions. Firstly, we used the original Gated SAE loss in Rajamanoharan et al. (2024), which uses a L1 sparsity penalty and requires resampling (Bricken et al., 2023) – periodic re-initialization of dead features – in order to train effectively. Secondly, we used a modified Gated SAE loss function that replaces the L1 sparsity penalty with the RI-L1 sparsity penalty described in Section 2; see Appendix D for details. With this modified loss function, we no longer need to use resampling to avoid dead features. We trained TopK SAEs using the AuxK auxiliary loss described in Gao et al. (2024) with $K_{\text{aux}} = 512$, which helps reduce the number of dead features. We also used an approximate algorithm for computing the top $K$ activations (Chern et al., 2022) after finding it produces similar results to exact TopK while being much faster (Appendix E). See Appendix I for further details of our training methodology and Appendix J for further details on the experiments described below.

### 5.1 EVALUATING THE SPARSITY-FIDELITY TRADE-OFF

**Methodology**  For a fixed SAE architecture and dictionary size, we trained SAEs of varying levels of sparsity by sweeping either the sparsity coefficient $\lambda$ (for JumpReLU or Gated SAEs) or $K$ (for TopK SAEs). We then plot curves showing, for each SAE architecture, the level of reconstruction fidelity attainable at a given level of sparsity.

**Metrics**  We use the mean L0-norm of feature activations, $\mathbb{E}_{\mathbf{x}} \|\mathbf{f}(\mathbf{x})\|_0$, as a measure of sparsity. To measure reconstruction fidelity, we use two metrics. Our primary metric is delta LM loss, the

---

[7]Intuitively, the first term in Eq. (10) measures the rate at which the expected reconstruction loss would increase if we increase $\theta_i$ – thereby pushing a small number of features that are currently used for reconstruction below the updated threshold. Similarly, the second term is $-\lambda$ multiplied by the rate at which the mean number of features used for reconstruction (i.e. mean L0) would *decrease* if we increase the threshold $\theta_i$. The density $p_i(\theta_i)$ comes into play because impact of a small change in $\theta_i$ on either the reconstruction loss or sparsity depends on how often feature activations occur very close to the current threshold.

[8]We did not include ProLU SAEs (Taggart, 2024) in our comparisons, despite their similarities to JumpReLU SAEs, because prior work has established that ProLU SAEs do not produce as faithful reconstructions as Gated or TopK SAEs at a given sparsity level (Gao et al., 2024).

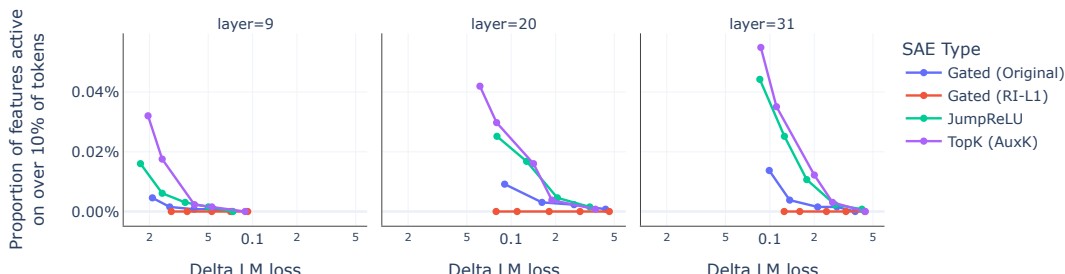

Figure 4: The proportion of features that activate very frequently versus delta LM loss by SAE type for Gemma 2 9B residual stream SAEs. TopK and JumpReLU SAEs tend to have relatively more very high frequency features – those active on over 10% of tokens – than Gated SAEs. If we instead count features that are active on over 1% of tokens (Fig. 17), the picture is more mixed: Gated SAEs can have more of these high (but not necessarily very high) features than JumpReLU SAEs, particularly in the low loss (and therefore lower sparsity) regime. A log scale is used on the $x$-axis.

increase in the cross-entropy loss experienced by the LM when we splice the SAE into the LM's forward pass. As a secondary metric, we also present in Fig. 16 curves that use fraction of variance unexplained (FVU) – also called the normalized loss (Gao et al., 2024) as a measure of reconstruction fidelity. This is the mean reconstruction loss $\mathcal{L}_{\text{reconstruct}}$ of a SAE normalized by the reconstruction loss obtained by always predicting the dataset mean.

**Results** Fig. 2 compares the sparsity-fidelity trade-off for JumpReLU, Gated and TopK SAEs trained on Gemma 2 9B residual stream activations. JumpReLU SAEs consistently offer similar or better fidelity at a given level of sparsity than TopK or Gated SAEs. Similar results are obtained for SAEs of each type trained on MLP or attention output activations, as shown in Fig. 14 and Fig. 15.

**Ablation study** In Appendix H.2 we present the results of an ablation study showing that *both* the JumpReLU activation function and the L0 sparsity penalty are necessary for obtaining this improvement in fidelity at fixed sparsity.

## 5.2 FEATURE ACTIVATION FREQUENCIES

For a given SAE, we are interested in both the proportion of learned features that are active very frequently and the proportion of features that are almost never active ("dead" features). Prior work has found that TopK SAEs tend to have more high frequency features than Gated SAEs (Cunningham & Conerly, 2024), and that these features tend to be less interpretable when sparsity is also low.

Fig. 4 shows, for JumpReLU, Gated and TopK SAEs, how the fraction of high frequency features varies with SAE fidelity (as measured by delta LM loss). TopK and JumpReLU SAEs consistently have more high frequency features – features that activate on over 10% of tokens – than Gated SAEs, although the fraction drops close to zero for SAEs in the low fidelity / high sparsity regime. Across all layers and frequency thresholds, JumpReLU SAEs have either similar or fewer high frequency features than TopK SAEs. Finally, it is worth noting that in all cases the number of high frequency features remains low in proportion to the widths of these SAEs, with fewer than 0.06% of features activating more than 10% of the time even for the highest L0 SAEs. Fig. 18 compares the proportion of "dead" features – which we defined to be features that activate on fewer than one in $10^7$ tokens – between JumpReLU, Gated and TopK SAEs. Both JumpReLU SAEs and TopK SAEs (with the AuxK loss) consistently have few dead features, without needing resampling.

## 5.3 INTERPRETABILITY OF SAE FEATURES

Although the results above suggest that JumpReLU SAEs provide more faithful reconstructions at a given level of sparsity than Gated or TopK SAEs, we would like to confirm that this improvement does not come at the cost of JumpReLU features being less interpretable. In the following two sections we evaluate the interpretability of JumpReLU, Gated and TopK SAE features using both a blinded human rating study, and by obtaining automated ratings using a language model.

### 5.3.1 Manual Interpretability

**Methodology** Our experimental setup resembles the manual interpretability study methodology described in Rajamanoharan et al. (2024). Five human raters[9] were asked to rate features from 81 SAEs, 27 of each type (Gated, TopK and JumpReLU) trained across multiple sites, layers and levels of sparsity. Each rater was presented with a feature dashboard (see Fig. 12), providing summary information and activating examples from the full activation spectrum of a feature, and asked whether the feature was mostly monosemantic, with allowed answers being 'Yes', 'Maybe' and 'No'.

**Results** As shown in Fig. 5a, all three SAE varieties exhibit similar rating distributions, consistent with prior results comparing TopK and Gated SAEs (Cunningham & Conerly, 2024; Gao et al., 2024) and furthermore showing that JumpReLU SAEs are similarly interpretable, i.e. that JumpReLU SAEs do not sacrifice feature interpretability to obtain more faithful reconstructions.

### 5.3.2 Automated Interpretability

To complement the manual interpretability study, we also evaluate SAE features using a language model rater. This allows us to collect many more samples than in a manual study, although LM ratings may be of more variable quality. To automatically rate features, we follow the two-step approach introduced in Bills et al. (2023) for neurons, and later used for SAE features (Bricken et al., 2023; Lieberum, 2024; Cunningham et al., 2023), of first using the LM to generate an explanation for a given feature and then asking the LM to predict feature activations based on that explanation.

**Methodology** We used Gemini 1.5 Flash (Gemini Team, 2024) for explanation generation and activation simulation. In the first step, Flash was given a list of sequences that activate a given feature to different degrees, together with discretized activation values, and asked for a natural language explanation of the feature consistent with these activation values. In the second step, we presented Flash the same token sequences in a fresh context and asked it to predict activation values using the explanation it gave in the first step. We then computed the Pearson correlation between the simulated and ground truth activation values for each sequence. We evaluated 1,000 randomly sampled features from 154 SAEs taken from all three types (Gated, JumpReLU and TopK).

**Results** We fit a logistic regression model to measure the effect of SAE type and L0 on the ability of the Gemini Flash to accurately simulate activations based on its explanations.[10] We also include terms in the model to account for each SAE's site and layer, after finding simulation quality varies widely between sites and layers (Fig. 13). Upon fitting the model, we find that JumpReLU SAE features have 27% greater odds of being well-simulated by Gemini Flash than Gated SAE features (OR=1.27, 95% CI: 1.23–1.31, $p < 0.001$) and that JumpReLU SAE features have similar odds of being well-simulated as TopK SAE features (OR=0.98, 95% CI: 0.95–1.01). We also find that doubling L0 lowers the odds of simulatability by 7% (OR=0.93, 95% CI: 0.92–0.94, $p < 0.001$). See Appendix J.1.2 for more on the interpretation and limitations of these results.

## 5.4 Disentanglement

In this section we evaluate how well the sparse decomposition provided by a SAE allows us to *disentangle* two concepts, i.e. vary one in isolation from the other (Schölkopf et al., 2021; Huang et al., 2024). Specifically, we work in a factual recall setting similar to Nanda et al. (2023), where we aim to use the SAE decomposition to change Gemma 2 9B's representation of the sport played by 50 baseball players to basketball, without disrupting its knowledge of these players' years of birth.

**Methodology** Full details can be found in Appendix J.2.1. In summary, we selected 50 athletes whom Gemma 2 9B correctly identifies as baseball players and can confidently state their year of birth. For each SAE, we then searched for (i) a basketball feature and (ii) a baseball feature, such

---

[9]All human raters in this study are either authors of this paper or members of the same research group.

[10]For this analysis, we set a threshold of 0.9 on the Pearson correlations $\rho$ between simulated and actual activation sequences to determine whether activations were well-simulated or not. A threshold of $\rho > 0.9$ corresponds to $R^2 \gtrsim 81\%$ in a regression setting, indicating a good fit. We found that changing this threshold does not significantly affect the conclusions of this analysis (see Appendix J.1).

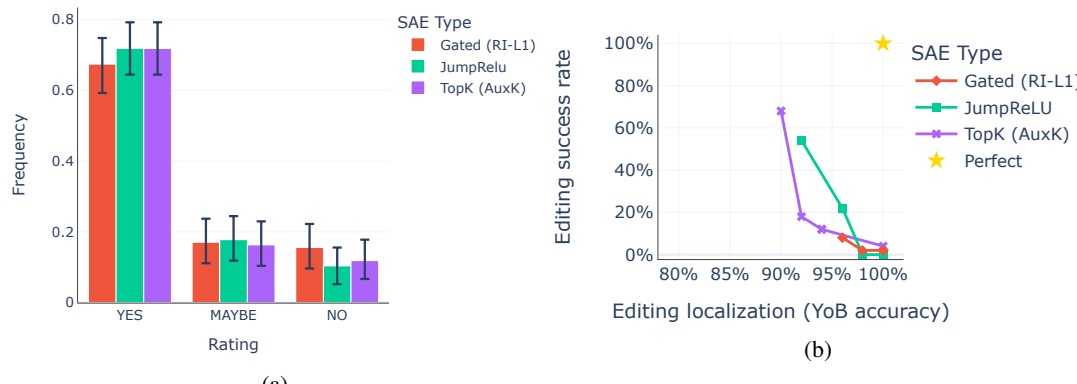

Figure 5: (a) Manual interpretability results (Section 5.3.1): features from all SAE architectures are rated as similarly interpretable by human raters. (b) Disentanglement results (Section 5.4): the trade-off between successfully updating baseball players' sport representations to basketball while avoiding disrupting the model's knowledge of their year of birth. To aid legibility, we omit points that are Pareto inferior to other points within each SAE type, i.e. only include points on each SAE type's Pareto frontier. A perfect intervention would receive a 100% score on both axes, as shown by the gold star annotated 'Perfect'.

that when we turn on the basketball feature and ablate the baseball feature on the tokens representing each athlete's name, the model would now report that the athlete plays basketball (not baseball), but still correctly report that athlete's year of birth. Where there were multiple candidate basketball or baseball features, we evaluated all combinations among the top 3 candidates for each feature.

**Results** Fig. 5b illustrates the trade-off between editing success – as measured by the proportion of the 50 athletes where the intervention successfully caused the model to report that they played basketball – versus editing localization – as measured by the proportion of athletes for which the model was still able to correctly report the year of birth despite the intervention. With JumpReLU and TopK SAEs, we were able to find single baseball and basketball features that successfully changed the sport representation of a majority of players, albeit with some disruption to the model's knowledge of these players' years of birth. Gated SAEs however performed poorly on this task, with none of the SAEs evaluated possessing features that were able to change the sport of more than 4 of the 50 athletes. See Appendix J.2.2 for further discussion of these results.

## 6 RELATED WORK

Recent interest in training SAEs on LM activations (Sharkey et al., 2022; Bricken et al., 2023; Cunningham et al., 2023) stems from the twin observations that many concepts appear to be linearly represented in LM activations (Elhage et al., 2021; Gurnee et al., 2023; Olah et al., 2020; Park et al., 2023) and that dictionary learning (Mallat & Zhang, 1993; Olshausen & Field, 1997) may help uncover these representations at scale. It is also hoped that the sparse representations learned by SAEs may be a better basis for identifying computational subgraphs that carry out specific tasks in LMs (Wang et al., 2023; Conmy et al., 2023; Dunefsky et al., 2024) and for finer-grained control over LMs' outputs (Conmy & Nanda, 2024; Templeton et al., 2024).

Recent improvements to SAE architectures – including TopK SAEs (Gao et al., 2024) and Gated SAEs (Rajamanoharan et al., 2024) – as well as improvements to initialization and sparsity penalties. Conerly et al. (2024) have helped ameliorate the trade-off between sparsity and fidelity and overcome the challenge of SAE features dying during training. Like JumpReLU SAEs, both Gated and TopK SAEs possess a thresholding mechanism that determines which features to include in a reconstruction; indeed, with weight sharing, Gated SAEs are mathematically equivalent to JumpReLU SAEs, although they are trained using a different loss function. JumpReLU SAEs are also closely related to ProLU SAEs (Taggart, 2024), which use a (different) STE to train an activation threshold, but do not match the performance of Gated or TopK SAEs (Gao et al., 2024).

The activation function defined in Eq. (2) was named JumpReLU in Erichson et al. (2019) and TRec in Konda et al. (2015). Both TopK and JumpReLU activation functions are closely related to activation pruning techniques such as ASH (Djurisic et al., 2023). The term *straight through estimator* was introduced in Bengio et al. (2013), although it is an old idea.[11] STEs have found applications in areas such as training quantized networks (e.g. Hubara et al. (2016)) and circumventing defenses to adversarial examples (Athalye et al., 2018). Our interpretation of STEs in terms of gradients of the expected loss is related to Yin et al. (2019), although they do not make the connection between STEs and KDE. Louizos et al. (2018) also show how it is possible to train models using a L0 sparsity penalty – on weights rather than activations in their case – by introducing stochasticity in the weights and taking the gradient of the expected loss.

## 7    CONCLUSION

Our evaluations show that JumpReLU SAEs produce reconstructions that consistently match or exceed the faithfulness of TopK SAEs, and exceed the faithfulness of Gated SAEs, at a given level of sparsity. They also show that the average JumpReLU SAE feature is similarly interpretable to the average Gated or TopK SAE feature, according to manual raters and automated evaluations. Although JumpReLU SAEs do have relatively more very high frequency features than Gated SAEs, they are similar to TopK SAEs in this respect.

In light of these observations, and taking into account the efficiency of training with the JumpReLU loss – which requires no auxiliary terms (unlike Gated SAEs) and does not involve relatively expensive TopK operations – we consider JumpReLU SAEs to be a mild improvement over existing SAE training methodologies.

**Limitations**    We note two key limitations with our study. Firstly, the evaluations presented in this paper concern training SAEs on several sites and layers of a single model, Gemma 2 9B. In Appendix G we also show that JumpReLU provides a better sparsity-vs-fidelity trade off than TopK and Gated SAEs when trained on Pythia 2.8B (Biderman et al., 2023) activations. Nevertheless we would welcome attempts to replicate our results on other model families. Secondly, since the main contribution of this paper is a methodology for successfully training SAEs with discontinuous loss functions, we have focused our evaluations on a few key metrics: the sparsity-fidelity trade-off, feature interpretability, feature activation frequencies, and disentanglement in a single setting. It would be valuable to compare these SAE varieties on a broader selection of metrics, particularly on evaluations that closely correspond to downstream tasks that may benefit from SAEs.

**Further work**    JumpReLU SAEs do suffer from a few limitations that we hope can be improved with further work:

- We suspect it may be possible to tweak the JumpReLU loss function further to tackle the appearance of high frequency features (Section 5.2).
- JumpReLU SAEs introduce new hyperparameters – the initial value of $\boldsymbol{\theta}$ and bandwidth parameter $\varepsilon$ – that require selecting. Although we found these to transfer reliably (with data normalization in place), there may be more principled ways to choose these hyperparameters, e.g. by adopting automatic bandwidth selection methods used for KDE.
- STEs can be used to train SAEs with other sparsity penalties that may prove more useful than L0. For example, in Appendix F, we train JumpReLU SAEs to attain a desired target level of sparsity $L_0^{\text{target}}$ by using the sparsity loss

$$\mathcal{L}_{\text{sparsity}}(\mathbf{x}) = \lambda \left( \|\mathbf{f}(\mathbf{x})\|_0 / L_0^{\text{target}} - 1 \right)^2, \tag{13}$$

  instead of an L0 penalty. STEs thus open up the possibility of training SAEs with other discontinuous loss functions that may further improve SAE quality or usability.

---

[11]Even the Perceptron learning algorithm (Rosenblatt, 1958) can be understood as using a STE to train through a step function discontinuity.

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

## A    DIFFERENTIATING INTEGRALS INVOLVING HEAVISIDE STEP FUNCTIONS

We start by reviewing some results about differentiating integrals (and expectations) involving Heaviside step functions.

**Lemma 1.** *Let $\mathbf{X}$ be a $n$-dimensional real random variable with probability density $p_{\mathbf{X}}$ and let $Y = g(\mathbf{X})$ for a differentiable function $g : \mathbb{R}^n \to \mathbb{R}$. Then we can express the probability density function of $Y$ as the surface integral*

$$p_Y(y) = \int_{\partial V(y)} p_{\mathbf{X}}(\mathbf{x}') \mathrm{d}S \tag{14}$$

*where $\partial V(y)$ is the surface $g(\mathbf{x}) = y$ and $\mathrm{d}S$ is its surface element.*

*Proof.* From the definition of a probability density function:

$$p_Y(y) := \frac{\partial}{\partial y} \mathbb{P}(Y < y) \tag{15}$$

$$= \frac{\partial}{\partial y} \int_{V(y)} p_{\mathbf{X}}(\mathbf{x}) \mathrm{d}^n \mathrm{x} \tag{16}$$

where $V(y)$ is the volume $g(\mathbf{x}) < y$. Eq. (14) follows from an application of the multidimensional Leibniz integral rule. □

**Theorem 1.** *Let $\mathbf{X}$ and $y$ once again be defined as in Lemma 1. Also define*

$$A(y) := \mathbb{E}\left[f(\mathbf{X})H(g(\mathbf{X}) - y)\right] \tag{17}$$

*where $H$ is the Heaviside step function for some function $f : \mathbb{R}^n \to \mathbb{R}$. Then, as long as $f$ is differentiable on the surface $g(\mathbf{x}) = y$, the derivative of $A$ at $y$ is given by*

$$A'(y) = -\mathbb{E}\left[f(\mathbf{X})|Y = y\right] p_Y(y) \tag{18}$$

*Proof.* We can express $A(y)$ as the volume integral

$$A(y) = \int_{V(y)} f(\mathbf{x}) p_{\mathbf{X}}(\mathbf{x}) \mathrm{d}^n \mathbf{x} \tag{19}$$

where $V(y)$ is now the volume $g(\mathbf{x}) > y$. Applying the multidimensional Leibniz integral rule (noting that $f$ is differentiable on the boundary of $V(y)$, we therefore obtain

$$A'(y) = -\int_{\partial V(y)} f(\mathbf{x}) p_{\mathbf{X}}(\mathbf{x}) \mathrm{d}S \tag{20}$$

where $\partial V$ is the surface $g(\mathbf{x}) = y$. Eq. (18) follows by noting that $p_{\mathbf{X}}(\mathbf{x}) = p_{\mathbf{X}|Y=y}(\mathbf{x}) p_Y(y)$ and thus substituting Eq. (14) into Eq. (20). □

**Lemma 2.** *With the same definitions as in Theorem 1, the expected value*

$$B(y) := \mathbb{E}\left[f(\mathbf{X})H(g(\mathbf{X}) - y))^2\right], \tag{21}$$

*which involves the square of the Heaviside step function, is equal to $A(y)$.*

*Proof.* Expressed in integral form, both $A(y)$ and $B(y)$ have the same domains of integration (the volume $g(\mathbf{x}) > y$) and integrands; therefore their values are identical. □

## B  DIFFERENTIATING THE EXPECTED LOSS

The JumpReLU loss is given by

$$\mathcal{L}_{\boldsymbol{\theta}}(\mathbf{x}) := \|\mathbf{x} - \hat{\mathbf{x}}(\mathbf{f}(\mathbf{x}))\|_2^2 + \lambda \|\mathbf{f}(\mathbf{x})\|_0. \tag{6}$$

By substituting in the following expressions for various terms in the loss:

$$f_i(\mathbf{x}) = \pi_i(\mathbf{x})H(\pi_i(\mathbf{x}) - \theta_i), \tag{22}$$

$$\hat{x}(\mathbf{f}) = \sum_{i=1}^{M} f_i(\mathbf{x})\mathbf{d}_i + \mathbf{b}_{\text{dec}}, \tag{23}$$

$$\|\mathbf{f}(\mathbf{x})\|_0 = \sum_{i=1}^{M} H(\pi_i(\mathbf{x}) - \theta_i), \tag{24}$$

taking the expected value, and differentiating (making use of the results of the previous section), we obtain

$$\frac{\partial \mathbb{E}_{\mathbf{x}}\left[\mathcal{L}_{\boldsymbol{\theta}}(\mathbf{x})\right]}{\partial \theta_i} = \left(\mathbb{E}_{\mathbf{x}}\left[J_i(\mathbf{x})|\pi_i(\mathbf{x}) = \theta_i\right] - \lambda\right)p_i(\theta_i) \tag{25}$$

where $p_i$ is the probability density function for the pre-activation $\pi_i(\mathbf{x})$ and

$$J_i(\mathbf{x}) := 2\theta_i \mathbf{d}_i \cdot \left[\mathbf{x} - \mathbf{b}_{\text{dec}} - \tfrac{1}{2}\theta_i \mathbf{d}_i - \sum_{j \neq i}^{M} \pi_j(\mathbf{x})\mathbf{d}_j H(\pi_j(\mathbf{x}) - \theta_j)\right]. \tag{26}$$

We can express this derivative in the more succinct form given in Eq. (10) and Eq. (11) by defining

$$I_i(\mathbf{x}) := 2\theta_i \mathbf{d}_i \cdot [\mathbf{x} - \hat{\mathbf{x}}(\mathbf{f}(\mathbf{x}))] \tag{27}$$

$$= 2\theta_i \mathbf{d}_i \cdot \left[\mathbf{x} - \mathbf{b}_{\text{dec}} - \sum_{j=1}^{M} \pi_j(\mathbf{x})\mathbf{d}_j H(\pi_j(\mathbf{x}) - \theta_j)\right].$$

and adopting the convention $H(0) := \frac{1}{2}$; this means that $I_i(\mathbf{x}) = J_i(\mathbf{x})$ whenever $\pi_i(\mathbf{x}) = \theta_i$, allowing us to replace $J_i$ by $I_i$ within the conditional expectation in Eq. (25).

## C  USING STEs TO PRODUCE A KERNEL DENSITY ESTIMATOR

Using the chain rule, we can differentiate the JumpReLU loss function to obtain the expression

$$\frac{\partial \mathcal{L}_{\boldsymbol{\theta}}(\mathbf{x})}{\partial \theta_i} = -\left(\frac{I_i(\mathbf{x})}{\theta_i}\right)\frac{\partial}{\partial \theta_i}\text{JumpReLU}_{\theta_i}(\pi_i(\mathbf{x})) + \lambda\frac{\partial}{\partial \theta_i}H(\pi_i(\mathbf{x}) - \theta_i) \tag{28}$$

where $I_i(\mathbf{x})$ is defined as in Eq. (11). If we replace the partial derivatives in Eq. (28) with the pseudo-derivatives defined in Eq. (8) and Eq. (9), we obtain the following expression for the pseudo-gradient of the loss:

$$\frac{\eth \mathcal{L}_{\boldsymbol{\theta}}(\mathbf{x})}{\eth \theta_i} = \frac{I_i(\mathbf{x}) - \lambda}{\varepsilon}K\left(\frac{\pi_i(\mathbf{x}) - \theta_i}{\varepsilon}\right). \tag{29}$$

Computing this pseudo-gradient over a batch of observations $\mathbf{x}_1, \mathbf{x}_2, \ldots, \mathbf{x}_N$ and taking the mean, we obtain the kernel density estimator

$$\frac{1}{N\varepsilon}\sum_{\alpha=1}^{N}\left(I_i(\mathbf{x}_\alpha) - \lambda\right)K\left(\frac{\pi_i(\mathbf{x}_\alpha) - \theta_i}{\varepsilon}\right). \tag{12}$$

## D  COMBINING GATED SAEs WITH THE RI-L1 SPARSITY PENALTY

Gated SAEs compute two encoder pre-activations:

$$\boldsymbol{\pi}_{\text{gate}}(\mathbf{x}) := \mathbf{W}_{\text{gate}}\mathbf{x} + \mathbf{b}_{\text{gate}}, \tag{30}$$

$$\boldsymbol{\pi}_{\text{mag}}(\mathbf{x}) := \mathbf{W}_{\text{mag}}\mathbf{x} + \mathbf{b}_{\text{mag}}. \tag{31}$$

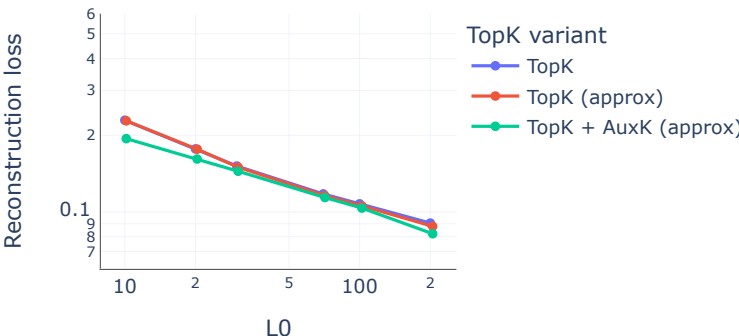

Figure 6: Using an approximation of TopK leads to similar performance as exact TopK. Adding the AuxK term to the loss function slightly improves fidelity at a given level of sparsity.

The first of these is used to determine which features are active, via a Heaviside step activation function, whereas the second is used to determine active features' magnitudes, via a ReLU step function:

$$\mathbf{f}_{\text{gate}}(\mathbf{x}) := H(\boldsymbol{\pi}_{\text{gate}}(\mathbf{x})) \tag{32}$$

$$\mathbf{f}_{\text{mag}}(\mathbf{x}) := \text{ReLU}(\boldsymbol{\pi}_{\text{mag}}(\mathbf{x})). \tag{33}$$

The encoder's overall output is given by the elementwise product $\mathbf{f}(\mathbf{x}) := \mathbf{f}_{\text{gate}}(\mathbf{x}) \odot \mathbf{f}_{\text{mag}}(\mathbf{x})$. The decoder of a Gated SAE takes the standard form

$$\hat{\mathbf{x}}(\mathbf{f}) := \mathbf{W}_{\text{dec}}\mathbf{f} + \mathbf{b}_{\text{dec}}. \tag{34}$$

As in Rajamanoharan et al. (2024), we tie the weights of the two encoder matrices, parameterizing $\mathbf{W}_{\text{mag}}$ in terms of $\mathbf{W}_{\text{gate}}$ and a vector-valued rescaling parameter $\mathbf{r}_{\text{mag}}$:

$$(\mathbf{W}_{\text{mag}})_{ij} := (\exp(\mathbf{r}_{\text{mag}}))_i (\mathbf{W}_{\text{gate}})_{ij}. \tag{35}$$

The loss function used to train Gated SAEs in Rajamanoharan et al. (2024) includes a L1 sparsity penalty and auxiliary loss term, both involving the positive elements of $\boldsymbol{\pi}_{\text{gate}}$, as follows:

$$\mathcal{L}_{\text{gate}} := \|\mathbf{x} - \hat{\mathbf{x}}(\mathbf{f}(\mathbf{x}))\|_2^2 + \lambda \|\text{ReLU}(\boldsymbol{\pi}_{\text{gate}}(\mathbf{x}))\|_1 + \|\mathbf{x} - \hat{\mathbf{x}}_{\text{frozen}}(\text{ReLU}(\boldsymbol{\pi}_{\text{gate}}(\mathbf{x})))\|_2^2 \tag{36}$$

where $\hat{\mathbf{x}}_{\text{frozen}}$ is a frozen copy of the decoder, so that $\mathbf{W}_{\text{dec}}$ and $\mathbf{b}_{\text{dec}}$ do not receive gradient updates from the auxiliary loss term.

For our JumpReLU evaluations in Section 5, we also trained a variant of Gated SAEs where we replace the L1 sparsity penalty in Eq. (36) with the reparameterization-invariant L1 (RI-L1) sparsity penalty $S_{\text{RI-L1}}$ defined in Section 2, i.e. by making the replacement $\|\text{ReLU}(\boldsymbol{\pi}_{\text{gate}}(\mathbf{x})\|_1 \rightarrow S_{\text{RI-L1}}(\boldsymbol{\pi}_{\text{gate}}(\mathbf{x}))$, as well as unfreezing the decoder in the auxiliary loss term. As demonstrated in Fig. 2, Gated SAEs trained this way have a similar sparsity-vs-fidelity trade-off to SAEs trained using the original Gated loss function, without the need to use resampling to avoid the appearance of dead features during training.

## E  APPROXIMATING TOPK

We used the approximate TopK approximation `jax.lax.approx_max_k` (Chern et al., 2022) to train the TopK SAEs used in the evaluations in Section 5. Furthermore, we included the AuxK auxiliary loss function to train these SAEs. Supporting these decisions, Fig. 6 shows:

- That SAEs trained with an approximate TopK activation function perform similarly to those trained with an exact TopK activation function;
- That the AuxK loss slightly improves reconstruction fidelity at a given level of sparsity.

## F  TRAINING JUMPRELU SAES TO MATCH A DESIRED LEVEL OF SPARSITY

Using the same pseudo-derivatives defined in Section 3 it is possible to train JumpReLU SAEs with other loss functions. For example, it may be desirable to be able to target a specific level of sparsity

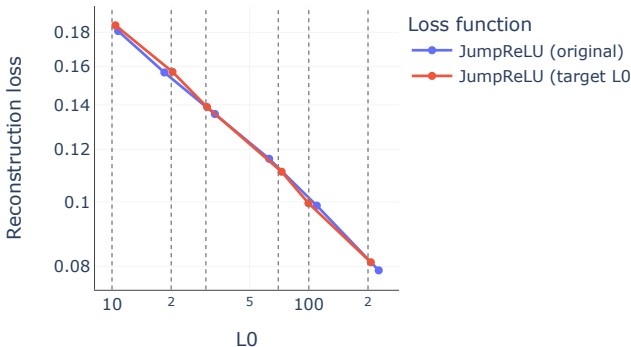

Figure 7: By using the sparsity penalty in Eq. (37), we can train JumpReLU SAEs to minimize reconstruction loss while maintaining a desired target level of sparsity. The vertical dashed grey lines indicate the target L0 values used to train the SAEs represented by the red dots closest to each line. These SAEs were trained setting $\lambda = 1$.

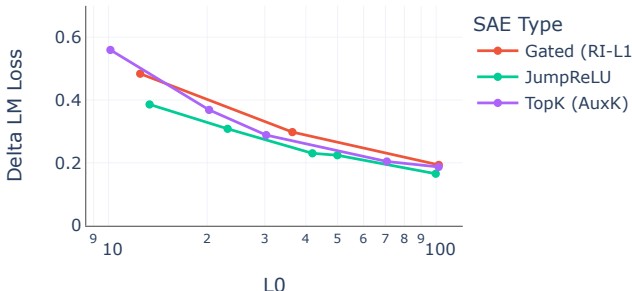

Figure 8: Reconstruction fidelity versus sparsity for JumpReLU, Gated and TopK SAEs trained on residual stream activations after layer 15 from Pythia 2.8B. We see a similar ordering of SAEs in terms of reconstruction fidelity at fixed sparsity as for the Gemma 2 9B SAEs evaluated in the rest of this paper, with JumpReLU SAEs providing more faithful reconstructions at a given level of sparsity than the other two architectures.

during training – as is possible by setting $K$ when training TopK SAEs – instead of the sparsity of the trained SAE being an implicit function of the sparsity coefficient and reconstruction loss.

A simple way to achieve this is by training JumpReLU SAEs with the loss

$$\mathcal{L}(\mathbf{x}) := \|\mathbf{x} - \hat{\mathbf{x}}(\mathbf{f}(\mathbf{x}))\|_2^2 + \lambda \left( \frac{\|\mathbf{f}(\mathbf{x})\|_0}{L_0^{\text{target}}} - 1 \right)^2. \tag{37}$$

Training SAEs with this loss on Gemma 2 9B's residual stream after layer 20, we find a similar fidelity-to-sparsity relationship to JumpReLU SAEs trained with the loss in Eq. (6), as shown in Fig. 7. Moreover, by using with the above loss, we are able to train SAEs that have L0s at convergence that are close to their targets, as shown by the proximity of the red dots in the figure to their respective vertical grey lines.

# G  COMPARING JUMPRELU, TOPK AND GATED SAES ON PYTHIA 2.8B

Appendix G compares fidelity versus sparsity for JumpReLU, TopK and Gated SAEs trained on middle layer (post-layer 15) residual stream activations from Pythia 2.8B. Results are qualitatively similar to those for Gemma 2 9B SAEs (Figs. 2 and 16): JumpReLU obtains better reconstruction fidelity at fixed sparsity than the other two architectures.

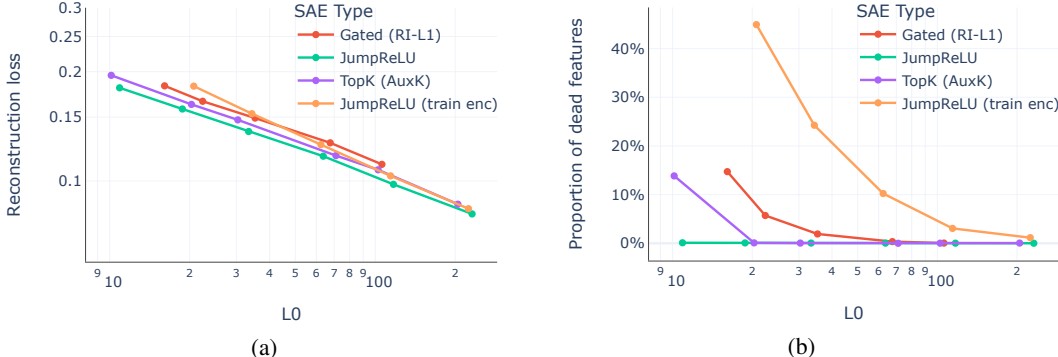

(a)                                    (b)

Figure 9: If we train JumpReLU SAEs by providing an additional gradient signal from the step functions in the loss to the encoder parameters as described in Appendix H.1 (orange), we obtain (a) worse reconstruction fidelity and (b) a significant proportion of dead features compared to the training approach described in the main body of the paper (green). These curves are for Gemma 2 9B post-layer 20 residual stream SAEs trained on 2B tokens.

## H    ABLATIONS

### H.1    PROVIDING PSEUDO-DERIVATIVES WITH RESPECT TO THE SAE ENCODER PARAMETERS

In principle we could define additional pseudo-derivatives so that the encoder parameters (i.e. not just the threshold parameters) also receive a gradient signal from the JumpReLU activation function and L0 loss. Concretely, this requires defining the following two pseudo-derivatives in addition to the pseudo-derivatives defined in Eqs. (8) and (9):

$$\frac{\eth}{\eth z}\text{JumpReLU}_\theta(z) := H(z - \theta) + \frac{\theta}{\varepsilon}K\left(\frac{z - \theta}{\varepsilon}\right) \tag{38}$$

$$\frac{\eth}{\eth z}H(z - \theta) := \frac{1}{\varepsilon}K\left(\frac{z - \theta}{\varepsilon}\right). \tag{39}$$

Note that these additional pseudo-derivatives are with respect to $z$, not $\theta$.

In practice however, we find that training JumpReLU SAEs this way leads to a notable drop in performance, as shown in Fig. 9a. We also observe, as shown in Fig. 9b, that JumpReLU SAEs trained this way exhibit a significant proportion of dead features, whereas JumpReLU SAEs trained without gradients to the encoder (i.e. using the approach described in Section 3) have hardly any dead features.

We conjecture that these dead features and drop in performance occur for the same reason that dead features occur in ReLU and Gated SAEs (without the RI-L1 penalty). When the encoder parameters receive a gradient signal from the sparsity penalty, this causes them to deactivate infrequently; once features activate too infrequently, this can lead to an irreversible loss of training signal, from which these features never recover.[12]

As a result, our recommended approach to training JumpReLU SAEs, as described in the main body of the paper (Section 3), is to only provide gradients to the threshold and not the encoder parameters. There is an interesting parallel to be made between training JumpReLU SAEs in this way and how TopK SAEs are trained: in both cases, the encoder and decoder parameters are trained to optimize a pure reconstruction loss (without any sparsity penalty). In the case of JumpReLU, the threshold parameters are trained using a sparsity penalty, whereas in TopK, the threshold is in effect dynamically chosen by the TopK operation. We suspect that the reason JumpReLU SAEs (trained with only a gradient to the threshold) and TopK SAEs suffer from few dead features is closely

---

[12]It is conceivable that – just as changing the L1 sparsity penalty to the RI-L1 ameliorates the dead features problem for ReLU and Gated SAEs – a similar tweak to the sparsity penalty used to train JumpReLU SAEs could also prevent features from dying during training even with a gradient signal to the encoder parameters. However, we leave it to further work to find such a modified sparsity penalty.

related to the fact that the encoder parameters for these two training methods are trained purely to reconstruct input activations, i.e. without these parameters directly receiving a training signal from a sparsity penalty.

### H.2 BOTH THE JUMPRELU ACTIVATION FUNCTION AND L0 SPARSITY PENALTY ARE NECESSARY FOR JUMPRELU SAES' IMPROVED PERFORMANCE

The way we define and train JumpReLU SAEs in this work modifies standard (i.e. ReLU) SAE training in two key respects: (a) we switch the encoder activation function from ReLU to JumpReLU and (b) we use a L0, not L1, sparsity penalty during training. In this section, we investigate whether the improved fidelity at given sparsity exhibited by JumpReLU SAEs is solely attributable to just one of these changes.

To do this, we train two SAE variants that incorporate, in isolation, just one of the changes listed in the previous paragraph:

- **ReLU SAE with L0 sparsity penalty.** We train a SAE with a ReLU activation function using a L0 penalty instead of the standard L1 penalty. In order for this to work – and considering that the ReLU activation function does not have a trainable threshold – we define straight-through-estimators as in Eqs. (38) and (39) so that the L0 penalty provides a gradient signal to the ReLU SAE's encoder parameters.
- **JumpReLU SAE with L1 sparsity penalty.** We also train a SAE using the L1 penalty but with a JumpReLU (instead of ReLU) activation function. With a JumpReLU activation function, there are two plausible ways to define the L1 sparsity penalty: we could either define it in terms of the L1-norm of the pre-activations, i.e. $\mathcal{L}_{\text{sparsity}}(\mathbf{x}) := \|\boldsymbol{\pi}(\mathbf{x})\|_1$, or in terms of the L1-norm of the difference between the pre-activations and their corresponding thresholds, i.e. $\mathcal{L}_{\text{sparsity}}(\mathbf{x}) := \|\boldsymbol{\pi}(\mathbf{x}) - \boldsymbol{\theta}\|_1$.[13] In practice, we find that the second definition – penalizing the positive difference between feature pre-activations and the corresponding feature thresholds – leads to significantly better performance, hence this is the version of the L1 sparsity penalty that we use in this study.

As shown in Fig. 10, both these SAE variants fare poorly when compared to JumpReLU SAEs trained with a L0 sparsity penalty as described in Section 3. Indeed, training a ReLU SAE with L0 sparsity penalty leads to significantly worse performance than even Gated SAEs, underscoring the importance of incorporating a thresholding mechanism in the SAE architecture as explained in Fig. 1.

Thus, we find that both the JumpReLU activation function and the L0 sparsity penalty introduced in the JumpReLU training methodology described in Section 3 are necessary for JumpReLU SAEs to attain the superior reconstruction fidelity versus sparsity curves shown in Figs. 2, 14 and 15.

### H.3 USING OTHER KERNEL FUNCTIONS

As described in Section 3, we used a simple rectangle function as the kernel, $K(z)$, within the pseudo-derivatives defined in Eq. (8) and Eq. (9). As shown in Fig. 11, similar results can be obtained with other common KDE kernel functions; there does not seem to be any obvious benefit to using a higher order kernel.

## I FURTHER DETAILS ON OUR TRAINING METHODOLOGY

- We normalize LM activations so that they have mean squared L2 norm of one during SAE training. This helps to transfer hyperparameters between different models, sites and layers.
- Except where noted, all SAEs used in the experiments in this paper were trained over 8 billion tokens.
- We trained all our SAEs with a learning rate of $7 \times 10^{-5}$ and batch size of 4,096.

---

[13]These two definitions become identical in the limit of the JumpReLU threshold becoming zero, i.e. in the limit that the JumpReLU activation function becomes a ReLU activation function. Hence, it is an empirical question which of these is the 'correct' generalization of the L1 penalty used to train ReLU SAEs.

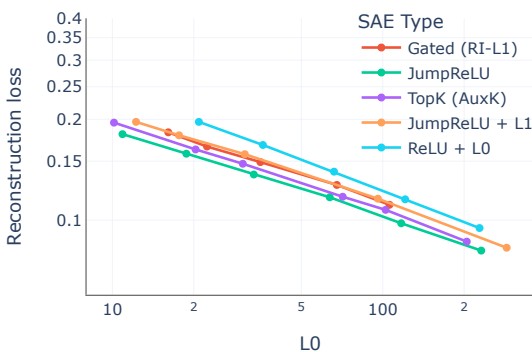

Figure 10: The two ablations described in Appendix H.2 – training a JumpReLU SAE with L1 sparsity penalty (orange) and training a ReLU SAE with L0 sparsity penalty (cyan) – obtain worse fidelity at given sparsity compared to the JumpReLU SAE training methodology introduced in Section 3. These curves are for Gemma 2 9B post-layer 20 residual stream SAEs trained on 2B tokens.

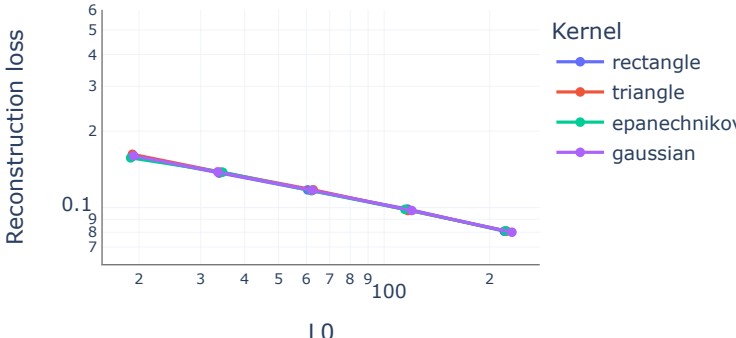

Figure 11: Using different kernel functions to compute the pseudo-derivatives defined in Eq. (8) and Eq. (9) has little impact on fidelity-vs-sparsity curves. These curves are for Gemma 2 9B post-layer 20 residual stream SAEs trained on 2B tokens.

- As in Rajamanoharan et al. (2024), we warm up the learning rate over the first 1,000 steps (4M tokens) using a cosine schedule, starting the learning rate at 10% of its final value (i.e. starting at $7 \times 10^{-6}$).
- We used the Adam optimizer (Kingma & Ba, 2017) $\beta_1 = 0$, $\beta_2 = 0.999$ and $\epsilon = 10^{-8}$. In our initial hyperparameter study, we found training with lower momentum ($\beta_1 < 0.9$) produced slightly better fidelity-vs-sparsity carves for JumpReLU SAEs, although differences were slight.
- We use a pre-encoder bias during training (Bricken et al., 2023) – i.e. subtract $\mathbf{b}_{\text{dec}}$ from $\mathbf{x}$ prior to the encoder. Through ablations we found this to either have no impact or provide a small improvement to performance (depending on model, site and layer).
- For JumpReLU SAEs we initialized the threshold $\boldsymbol{\theta}$ to 0.001 and the bandwidth $\varepsilon$ also to 0.001. These parameters seem to work well for a variety of LM sizes, from single layer models up to and including Gemma 2 9B.
- For Gated RI-L1 SAEs we initialized the norms of the decoder columns $\|\mathbf{d}_i\|_2$ to 0.1.
- We trained all SAEs except for Gated RI-L1 while constraining the decoder columns $\|\mathbf{d}_i\|_2$ to 1.[14]
- Following Conerly et al. (2024) we set $\mathbf{W}_{\text{enc}}$ to be the transpose of $\mathbf{W}_{\text{dec}}$ at initialization (but thereafter left the two matrices untied) when training of all SAE types, and warmed up $\lambda$ linearly over the first 10,000 steps (40M tokens) for all except TopK SAEs.

---

[14]This is not strictly necessary for JumpReLU SAEs and we subsequently found that training JumpReLU SAE without this constraint does not change fidelity-vs-sparsity curves, but we have not fully explored the consequences of turning this constraint off.

**Activations in range 0.0756 to 0.0882**

2 5 5 ; \n                  ctx . put ImageData ( imageData , x , y ); \n                } else { \n                  var offset = 4 * **y** * this . str (), sec _ addr , sec _ size ); \n \n                if ( sec _ addr != 0 ) \n                { \n                ADDR INT high _**addr** = sec _ 0 ); else if ( this . allocated > value ) this . _n me Resize Buffer ( this . allocated = value ); \n                this . length = **value** ; \n { \n                if ( SEC _ Is Executable ( sec ) /* && SEC _ Name ( sec ) == ".text" */ ) \n                { \n                **ADDR** INT sec _ !(" Error : {}", err ); \n                return ; \n         } \n       }; \n \n   match cs . dis asm ( & buf [ 0 .. bytes ], **addr** ,   0 \n \n If parameters of the extent are incorrect, the command is terminated with Unit Check ( Command Reject ), Channel End and Device End status . \n \n If the **address** able block size self ) : \n       self . uint 3 2 ('id') \n       self . uint 3 2 ('address') \n       self . uint 3 2 ('**size**') \n ( err ) => panic !( err . to _ string ()), \n       }; \n \n   return buffer ; \n } \n \n fn make _ ram _ region ( **base** : u 6 tmp _ mem , tmp _ mem ); \n     for ( ssize _ t x = sizeof ( Instance Object ); x < cls -> instance _ size (); **x** += 8 : [ u 8 ; CODE _ SIZE ] = [ 0 ; CODE _ SIZE ]; \n   let bytes = vm :: read _ guest _ memory ( **addr** , & mut

**Activations in range 0.063 to 0.0756**

c _ str (), sec _ addr , sec _ size ); \n \n                if ( sec _ addr != 0 ) \n                { \n                ADDR INT high _**addr** = !( err . to _ string ()), \n     }; \n \n   return buffer ; \n } \n \n fn make _ ram _ region ( base : u 6 4 , **size** : usize \n { \n   let ram _ region = vm :: alloc _ memory _ region ( size ); \n     vm :: map _ memory _ region ( **base** , HV _ MEMORY         { \n             for ( b len = 0 ; b len < 1 6 ; b len ++) \n                b Dat abu ffer [ f **Addr** + b len , f ; \n       e [ t >> 5 ] |= 1 2 8 << **t** % 3 2 , e [ 1 4 + ( **t** + 6 Word ( cmd lst , ( byte ) s Uart I sp . Length , ref rev lst , ref b len ); \n \n     for ( ushort f **Addr** = 0 ;             for ( b len = 0 ; b len < 1 6 ; b len ++) \n                { \n                b Dat abu ffer [ f **Addr** + b len Transport :: read _ virt ( uint 8 _ t * buf , uint 3 2 _ t len ) \n { \n   assert ( m _ pos + len <= m  any data not already done by \n            * the caller and add in any partial checksum . \n                */ \n                if (( u _ char *) w > dp -> db ); \n \n            for ( ushort f **Addr** = 0 ; f **Addr** < Element Define . RAM _ MAX _ CAP ; f **Addr** += 1 6 ) \n            {

**Activations in range 0.0504 to 0.063**

extent of the area ( or space ) on the virtual disk for which subsequent commands are valid . \n \n   \item LOC ATE command to specify a specific **address** and the amount this -> as . write _ xor ( tmp _ mem , tmp _ mem ); \n   for ( ssize _ t x = sizeof ( Instance Object ) ; x < cls */ \n unsigned int \n ip _ ck sum ( mp , **offset** , sum ) \n     register m blk _ t * mp ; \n     register int **offset** ; \n this . readable Stream ); \n       } \n \n   async consume ToEnd (): Promise < void > { \n       this . pos += this . buffers Length ; **\n**                this . Explicit Scaling List Used ( false ); \n   } \n # endif \n   if ( pc Slice -> getFirst C tu Rs Addr In Slice () == **0** ) \n size ); \n \n            if ( sec _ addr != 0 ) \n            { \n                ADDR INT high _ addr = sec _ addr + sec _ **size** ; \n \n is used \n * when computing partial checksum s . \n * For non STRU IO _ IP m bl ks , assumes mp -> b _ rptr + **offset** is 1 Replace " for loop " with standard mem set if possible . \n */ \n static void MD 5 _ memset ( POINTER output , int value , unsigned int **len** ) { \n * the caller and add in any partial checksum . \n                */ \n                if (( u _ char *) w > dp -> db _ stru io base || \n < | BOS |> p Tmp Img Data , Img Info . m _ Width , Img Info . m _ Height , Color Channel Count , X * Copy Width , Y * Copy

Figure 12: An extract of one of the feature dashboards used by raters in the manual interpretability study. The dashboard shows examples of text where the feature being rated is active, across the full range of activations (only three such ranges are shown in this cropped screenshot). The token where the feature maximally activates is highlighted in bold, with the orange shading also visually indicating the strength of activation. By hovering over a token (not shown), the rater can see the exact value of the feature's activation at that token. The SAE feature in this example seems to activate on tokens relating to the addressing of RAM.

- We used resampling (Bricken et al., 2023) – periodically re-initializing the parameters corresponding to dead features – with Gated (original loss) SAEs, but did not use resampling with Gated RI-L1, TopK or JumpReLU SAEs.

## J    FURTHER EXPERIMENTAL DETAILS AND RESULTS

### J.1    INTERPRETABILITY STUDIES

#### J.1.1    MANUAL INTERPRETABILITY

For each site (attention output, MLP output, residual stream), each layer (9, 20, 31) and each architecture (Gated, TopK, JumpReLU) we picked three SAEs (from a sweep over multiple sparsity coefficients) that had L0 closest to 20, 75 and 150, for a total of 81 SAEs to study. Each rater rated a feature from every SAE, presented in a random order. Thus, in total we collected 405 samples, i.e. 5 per SAE. Fig. 12 shows an example of the dashboards used by raters in this study.

#### J.1.2    AUTOMATED INTERPRETABILITY

During the explanation step, activation sequences presented to Flash were binned and normalized to be integers between 0 and 10. We found that using a diverse few-shot prompt for both explanation generation and activation simulation was important for consistent results. 154 SAEs were used for this study, from all three types (Gated, JumpReLU and TopK), three layers (9, 20 and 31), three sites and five or six sparsity levels.

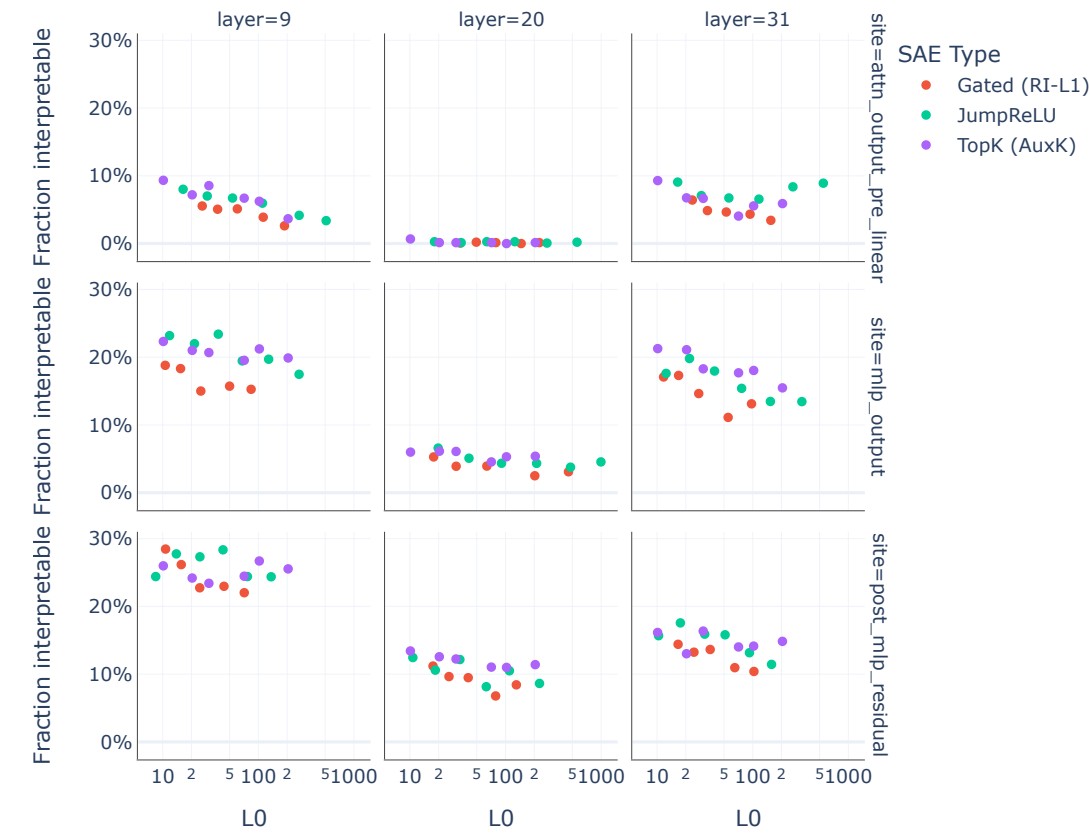

Figure 13: The fraction of features for which Gemini Flash was successfully able to simulate feature activations based on its explanations – where a successful simulation is deemed as one with a Pearson correlation $\rho > 0.9$ – varies widely between sites and layers. We adjust for this layer and site dependence when estimating the effect of SAE type and L0 on LM rated interpretability.

Table 1: Sensitivity analysis: dependence of odds ratios presented in Section 5.3.2 on the threshold used to label activations simulations as well-simulated.

| Threshold | Odds ratios | | |
|---|---|---|---|
| | Gated vs JumpReLU | TopK vs JumpReLU | Doubling L0 |
| 0.85 | 1.26 (CI: 1.22–1.30) | 0.98 (CI: 0.95–1.01) | 0.93 (CI: 0.92-0.94) |
| 0.90 | 1.27 (CI: 1.23–1.31) | 0.98 (CI: 0.95–1.01) | 0.93 (CI: 0.92-0.94) |
| 0.90 | 1.28 (CI: 1.23–1.34) | 0.98 (CI: 0.94–1.02) | 0.93 (CI: 0.92-0.94) |

Table 1 shows how the effect sizes obtained from the logistic regression model we fit to the results are sensitive to our choice of setting the threshold at which we deemed simulated activations to be well-simulated at $\rho > 0.9$. Lowering and raising this threshold does not change the qualitative picture.

Although the results of the automated interpretability study are interesting and consistent with the picture painted by the other evaluations in Section 5, we caution against reading too much into them. The fraction of SAE features that Gemini Flash was able to simulate well was less than 30% at any site or layer, and often lower than 10%, suggesting that there is a large gap between the quality of these automated ratings and the manual ratings of the previous section. Furthermore, it is plausible Flash is better able to simulate "simpler" features, that have more localized explanations (e.g. token-level features), which would put SAEs that are better able to identify complex or abstract features at a disadvantage.

## J.2 DISENTANGLEMENT EXPERIMENT

### J.2.1 DETAILED METHODOLOGY

We began by prompting Gemini 1.5 Pro (Gemini Team, 2024) to list 150 baseball players, 100 basketball players, 100 American football players and 100 ice hockey players. These sports were chosen on the basis that they are all team sports popular in North America, to minimize the risk of spurious correlations when searching for SAE features that represent these sports. We checked whether Gemma 2 9B could retrieve the sport played by each of these athletes by asking it to complete the prompt "Fact: the sportsperson <player-name> is known for playing the sport of"; on this basis we removed one football player that Gemma 2 9B could not correctly classify.

We then split this dataset into 50 baseball players who would be used for the disentanglement evaluation itself and the remaining baseball, basketball, football and ice hockey players who would be used to search for SAE features representing the sports of baseball and basketball. To perform this split, we first evaluated the probabilities assigned by Gemma 2 9B to four-digit completions of the prompt "Fact: the sportsperson <player-name> was born in the year " ranging from 1890 to 2019. The 50 players for whom Gemma 2 9B assigned the highest probabilities to their year of birth were selected for the disentanglement split, with the remaining players used for the feature search split.[15]

For each SAE in the experiment, we used the feature-search split of the dataset – consisting of 399 baseball, basketball, football and ice hockey players in almost equal proportion – to obtain two rank orderings of the SAE's features: firstly ordering the features by how well they classify baseball players and secondly ordering the features by how well they classify basketball players.

To do this, we first extracted SAE feature activations on the tokens representing each athlete's name in the prompt "Fact: the sportsperson <player-name>", hypothesizing that SAE activations at these name token positions are likely to contain features representing the sport played by that athlete (Meng et al., 2022; Geva et al., 2023; Nanda et al., 2023). We then took the sequence-wise maximum of the SAE activations across all name tokens for each player, yielding a scalar score per feature and per player. Finally, we computed the area under the ROC curve (AUROC) between each feature's scores for the athletes and their ground truth sport labels and used these AUROC values to rank order the features according to how well they separate baseball and non-baseball players (for the baseball feature ranking) or basketball and non-basketball players (for the basketball feature ranking).

Having done this, we discarded features that had an AUROC score below 0.75 or were not in the top-3 for each ranking, in order to get two lists of putative basketball and baseball features – each containing up to three features – for the SAE being evaluated.

For each combination of a basketball feature from the basketball features list and a baseball feature from the baseball features list, we would evaluate how well these two features are able to change the model's representation of the sport played by the athlete to basketball without affecting the model's knowledge of the athlete's year of birth. To perform this evaluation, we would compute the model's completions to the prompts "Fact: the sportsperson <player-name> is known for playing the sport of" and "Fact: the sportsperson <player-name> was born in the year " and use these completions to calculate the editing success and editing localization metrics shown in Fig. 5b. Crucially, we would compute the completions after *intervening* on the model, setting the chosen baseball feature's activation to zero and the chosen basketball feature's activation to the dataset median across the basketball players in the feature-search dataset.[16][17] Since our aim is to update the model's representations for the players in the prompts, we only apply this intervention at the tokens in each prompt that represent a player's name.

We evaluated 5 Gated (RI-L1) SAEs, 6 JumpReLU SAEs and 6 TopK SAEs using this procedure, all trained to reconstruct Gemma 2 9B's residual stream activations after layer 9; these are exactly

---

[15]The lowest birth year probability assigned to any of the players in the disentanglement split was 87%; i.e. the split contained players for whom the model confidently knows their years of birth.

[16]Unlike Makelov (2024), we use median ablation instead of activation patching. The principal reason for this is that it allows us to intervene at a variable number of name tokens straightforwardly; activation patching is non-trivial for non token-aligned datasets. However, we also believe our approach more realistically captures how practitioners may ultimately use SAEs in real-world model control tasks.

[17]When editing the LM's activations in this way, we found it important to include a reconstruction error term in our sparse decomposition as explained in Marks et al. (2024).

the same SAEs whose sparsity-fidelity metrics are shown in the left-hand facet of Fig. 2. We chose to evaluate layer 9 SAEs for this task since prior work has found that attributes a model knows about an entity are typically represented in early layers and extracted by attention heads at middle layers in the network (Geva et al., 2023; Nanda et al., 2023).

### J.2.2 FURTHER DISCUSSION OF THE RESULTS

Fig. 5b also appears to show that JumpReLU SAEs provide a better trade off between editing success and localization than TopK SAEs. However, we caution against drawing overly general conclusions about the disentanglement properties of JumpReLU versus TopK SAEs from these results due to some limitations of the methodology described above:

- Since we limit our interventions to those that involve a single basketball feature and single baseball feature, editing success depends crucially on whether those features are cleanly represented in the SAE being evaluated. Especially when evaluating on specialized concepts like a particular sport, it is important to evaluate on as many SAEs as possible (ideally multiple SAEs trained with the same hyperparameters but different seeds) in order to reduce the variance arising from the uncertainty in whether the features we wish to steer with are present or not.
- We have measured editing success and editing localization using just one prompt for each metric. Ideally, we should test the extent to which an intervention has changed the model's representation of the player's sport across multiple prompts. Similarly, we should test the extent to which the intervention has disrupted the model's knowledge of other facts about the player across both multiple facts (i.e. not just their year or birth) and multiple prompts to elicit each fact.
- Our evaluation measures disentanglement in a very particular setting. To draw general conclusions about the disentanglement properties of different SAE types, we would need to apply the methodology to multiple concepts, including concepts in non-factual settings.

Addressing these limitations will require both extra compute and extra effort to compile suitable datasets. Since the main contribution of this paper is to present a new SAE training methodology, rather than to make progress in the science of evaluating SAEs (which we nonetheless think is a very important research direction for the field), we leave these improvements to further work.

### J.3 EVALUATING THE SPARSITY-FIDELITY TRADE-OFF

All metrics for this evaluation, i.e. L0, delta LM loss and fraction of variance explained, were computed on 2,048 sequences of length 1,024, after excluding special tokens (pad, start and end of sequence) when aggregating the results. Fig. 14 and Fig. 15 plot reconstruction fidelity against sparsity for SAEs trained on Gemma 2 9B MLP and attention outputs at layers 9, 20 and 31. Fig. 16 uses fraction of variance explained (see Section 5) as an alternative measure of reconstruction fidelity, and again compares the fidelity-vs-sparsity trade-off for JumpReLU, Gated and TopK SAEs on MLP, attention and residual stream layer outputs for Gemma 2 9B layers 9, 20 and 31.

### J.4 FEATURE ACTIVATION FREQUENCIES

To measure activation frequencies, we collected SAE feature activation statistics over 10,000 sequences of length 1,024, and computed the frequency with which individual features fire on a randomly chosen token (excluding special tokens). Fig. 17 compares the number of features that activate on over 1% of tokens in JumpReLU, Gated and TopK SAEs. Fig. 19 compares feature activation frequency histograms for JumpReLU, TopK and Gated SAEs of comparable sparsity.

## K PSEUDO-CODE FOR IMPLEMENTING AND TRAINING JUMPRELU SAES

We include pseudo-code for implementing:

- The Heaviside step function with custom backward pass defined in Eq. (9).
- The JumpReLU activation function with custom backward pass defined in Eq. (8).
- The JumpReLU SAE forward pass.
- The JumpReLU loss function.

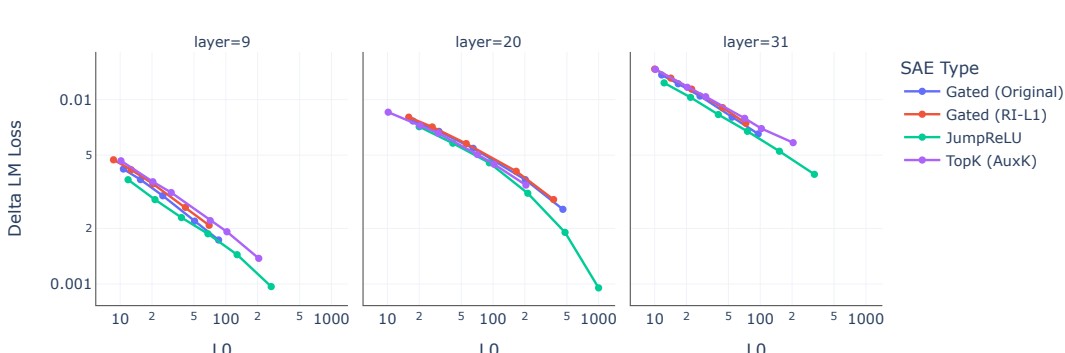

Figure 14: Comparing reconstruction fidelity versus sparsity for JumpReLU, Gated and TopK SAEs trained on Gemma 2 9B layer 9, 20 and 31 MLP outputs. JumpReLU SAEs consistently provide more faithful reconstructions (lower delta LM loss) at a given level of sparsity (as measured by L0).

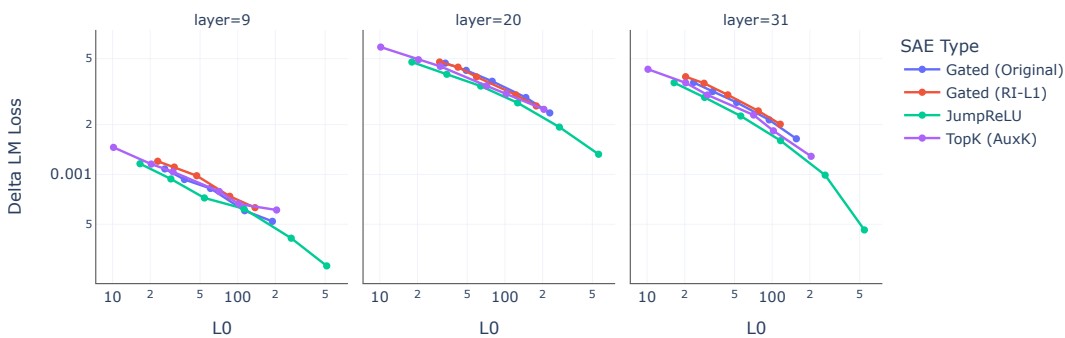

Figure 15: Comparing reconstruction fidelity versus sparsity for JumpReLU, Gated and TopK SAEs trained on Gemma 2 9B layer 9, 20 and 31 attention activations prior to the attention output linearity ($\mathbf{W}_O$). JumpReLU SAEs consistently provide more faithful reconstructions (lower delta LM loss) at a given level of sparsity (as measured by L0).

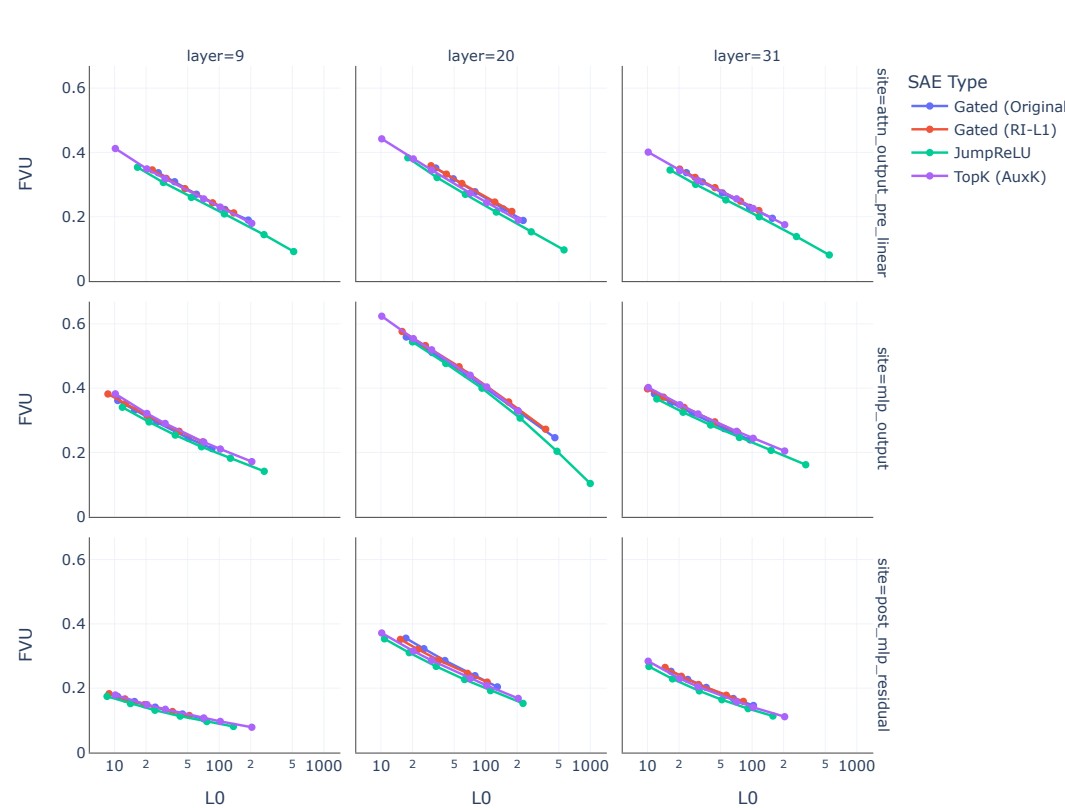

Figure 16: Comparing reconstruction fidelity versus sparsity for JumpReLU, Gated and TopK SAEs trained on Gemma 2 9B layer 9, 20 and 31 MLP, attention and residual stream activations using fraction of variance unexplained (FVU) as a measure of reconstruction fidelity.

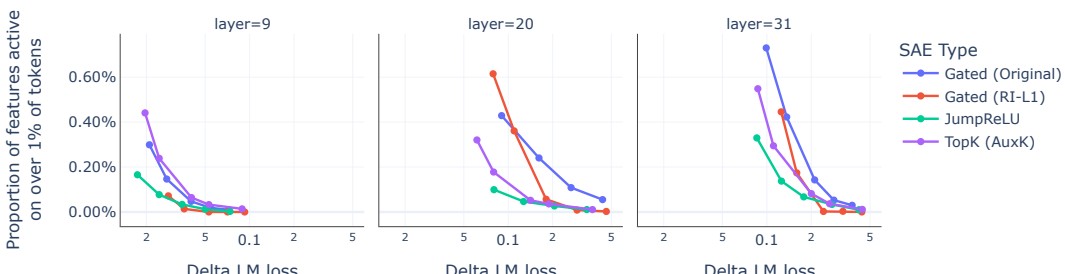

Figure 17: The proportion of features that activate on more than 1% of tokens versus delta LM loss by SAE type for Gemma 2 9B residual stream SAEs. Compared to the analogous plot for features that activate on more than 10% of tokens in Fig. 4, Gated SAEs can have more of these high (but not necessarily very high) features than JumpReLU SAEs, particularly in the low loss (and therefore lower sparsity) regime.

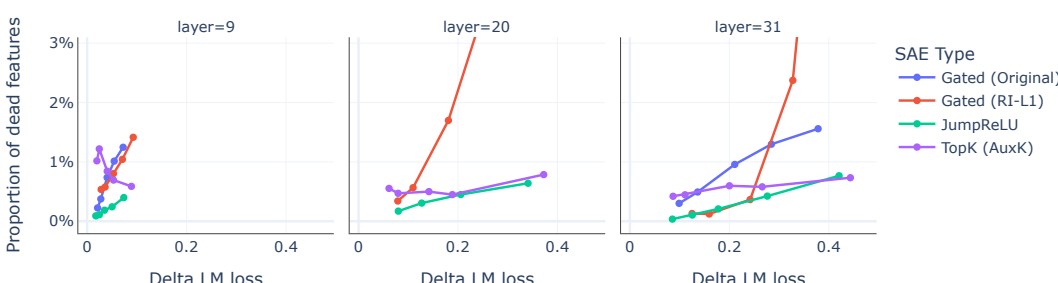

Figure 18: JumpReLU and TopK SAEs have few dead features (features that activate on fewer than one in $10^7$ tokens), even without resampling. Note that the original Gated loss (blue) – the only training method that uses resampling – had around 40% dead features at layer 20 and is therefore missing from the middle plot.

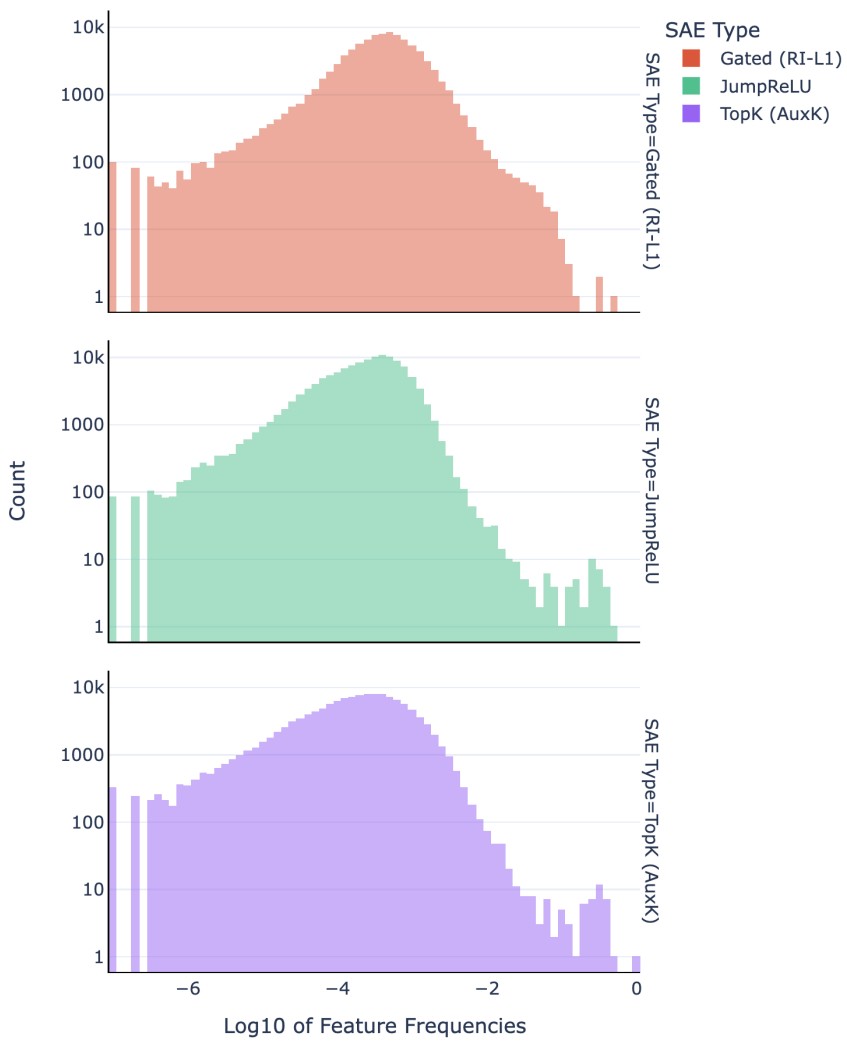

Figure 19: Feature frequency histograms for JumpReLU, TopK and Gated SAEs all with L0 approximately 70 (excluding features with zero activation counts). Note the log-scale on the y-axis: this is to highlight a small mode of high frequency features present in the JumpReLU and TopK SAEs. Gated SAEs do not have this mode, but do have a "shoulder" of features with frequencies between $10^{-2}$ and $10^{-1}$ not present in the JumpReLU and TopK SAEs.

Our pseudo-code most closely resembles how these functions can be implemented in JAX, but should be portable to other frameworks, like PyTorch, with minimal changes.

Two implementation details to note are:

- We use the logarithm of threshold, i.e. $\log(\boldsymbol{\theta})$, as our trainable parameter, to ensure that the threshold remains positive during training.
- Even with this parameterization, it is possible for the threshold to become smaller than half the bandwidth, i.e. that $\theta_i < \varepsilon/2$ for some $i$. To ensure that negative pre-activations can never influence the gradient computation, we take the ReLU of the pre-activations before passing these to the JumpReLU activation function or the Heaviside step function used to compute the L0 sparsity term. Mathematically, this has no impact on the forward pass (because pre-activations below the positive threshold are set to zero in both cases anyway), but it ensures that negative pre-activations cannot bias gradient estimates in the backward pass.

```python
def rectangle(x):
  return ((x > -0.5) & (x < 0.5)).astype(x.dtype)

### Implementation of step function with custom backward

@custom_vjp
def step(x, threshold):
  return (x > threshold).astype(x.dtype)

def step_fwd(x, threshold):
  out = step(x, threshold)
  cache = x, threshold # Saved for use in the backward pass
  return out, cache

def step_bwd(cache, output_grad):
  x, threshold = cache
  x_grad = 0.0 * output_grad # We don't apply STE to x input
  threshold_grad = sum(
      -(1.0 / bandwidth) * rectangle((x - threshold) / bandwidth) *
          output_grad,
      axis=0,
  )
  return x_grad, threshold_grad

step.defvjp(step_fwd, step_bwd)

### Implementation of JumpReLU with custom backward for threshold

@custom_vjp
def jumprelu(x, threshold):
  return x * (x > threshold)

def jumprelu_fwd(x, threshold):
  out = jumprelu(x, threshold)
  cache = x, threshold # Saved for use in the backward pass
  return out, cache

def jumprelu_bwd(cache, output_grad):
  x, threshold = cache
  x_grad = (x > threshold) * output_grad # We don't apply STE to x input
  threshold_grad = sum(
      -(threshold / bandwidth)
      * rectangle((x - threshold) / bandwidth)
      * output_grad,
      axis=0,
  )
  return x_grad, threshold_grad

jumprelu.defvjp(jumprelu_fwd, jumprelu_bwd)

### Implementation of JumpReLU SAE forward pass and loss functions

def sae(params, x, use_pre_enc_bias):
  # Optionally, apply pre-encoder bias
  if use_pre_enc_bias:
```

```
    x = x - params.b_dec

  # Encoder - see accompanying text for why we take the ReLU
  # of pre_activations even though it isn't mathematically
  # necessary
  pre_activations = relu(x @ params.W_enc + params.b_enc)
  threshold = exp(params.log_threshold)
  feature_magnitudes = jumprelu(pre_activations, threshold)

  # Decoder
  x_reconstructed = feature_magnitudes @ params.W_dec + params.b_dec

  # Also return pre_activations, needed to compute sparsity loss
  return x_reconstructed, pre_activations

### Implementation of JumpReLU loss

def loss(params, x, sparsity_coefficient, use_pre_enc_bias):
  x_reconstructed, pre_activations = sae(params, x, use_pre_enc_bias)

  # Compute per-example reconstruction loss
  reconstruction_error = x - x_reconstructed
  reconstruction_loss = sum(reconstruction_error**2, axis=-1)

  # Compute per-example sparsity loss
  threshold = exp(params.log_threshold)
  l0 = sum(step(pre_activations, threshold), axis=-1)
  sparsity_loss = sparsity_coefficient * l0

  # Return the batch-wise mean total loss
  return mean(reconstruction_loss + sparsity_loss, axis=0)
```

