# OpenReview forum: "Jumping Ahead: Improving Reconstruction Fidelity with JumpReLU Sparse Autoencoders"
_ICLR.cc/2025/Conference — Submitted to ICLR 2025_

### Official Review · Reviewer_EMQL · 2024-10-21

**Soundness:** 2
**Presentation:** 3
**Contribution:** 2
**Rating:** 3
**Confidence:** 2

**Summary:**

This paper aims to train sparse autoencoders (SAEs) to find causal features of language models, which is a problem in the field of mechanistic interpretability. This paper proposes to use the JumpReLU activation, illustrated in Fig. 3(a). The authors provide ways to estimate the gradient of the JumpReLU activation, as well as ways to estimate the gradient of $L_0$ norm regularization. The estimation mainly uses KDE.

**Strengths:**

1. The estimation of the $L_0$ norm's gradient makes sense and seems non-trivial to me.

**Weaknesses:**

1. The contribution seems quite incremental to me. The authors propose two things: (i) using the JumpReLU activation, and (ii) a way of estimating the gradient of JumpReLU and $L_0$ regularization. The JumpReLU activation is not invented in this work. And the estimation, though non-trivial, is derived from standard tools.

2. The experimental results show that the proposed method has similar performance to TopK SAE. The only plot where the proposed method seems better than TopK is Fig. 5(a), but this is actually rated by the authors themselves and people from the same group so it a bit questionable. TopK SAE (Gao et al.) was published on ArXiv on Jun 6, so is considered contemporaneous according to ICLR's guidelines. However, it's worth pointing out that the proposed method is not significantly better than TopK SAE.

3. One key ablation study is missing. The authors propose two things: JumpReLU, and using $L_0$ regularization instead of $L_1$. What will the performance be like if only one of them is used? For example, $L_0$ regularization + TopK? I imagine that this could have a very similar performance, which raises a question about the effectiveness of JumpReLU.

**Questions:**

1. What is the intuition behind JumpReLU? To me, TopK is very intuitive. However, I cannot understand why JumpReLU is reasonable. Is it derived from Gated SAE?

2. In line 86, you mentioned that TopK needs a partial sort while JumpReLU does not. However, what is the downside of this partial sort? Does it make the method slower? If so, how much slower? If it is only a bit slower, then it won't be a big issue.

3. Are there any qualitative experimental results? For example, can you provide some examples of the causal features you found using your method?

**Summary:** I am not an expert in mechanistic interpretability. To me, overall it seems that this paper is incremental and is not significantly better than a previous paper TopK SAE. Although TopK SAE is considered a contemporaneous work by ICLR's standard, this paper heavily depends on that paper so it is questionable. I feel that the contribution is not sufficient for acceptance at ICLR. My current rating is reject with low confidence, and I might change my score after discussing with my fellow reviewers and the AC.

---

> ### Author Response · Authors · 2024-11-21
>
> Thank you for your thoughtful and detailed feedback. We're particularly encouraged by your recognition of the non-trivial theoretical insights we introduce regarding training SAEs with parameter-discontinuous loss functions. Nevertheless, we appreciate your concerns and would like to address them directly.
>
> First, _regarding overall performance and significance_: Our evaluations show that JumpReLU SAEs consistently match or exceed the reconstruction fidelity of both Gated and TopK SAEs across all tested combinations of layers, sites and models (including Pythia 2.8B results in the Appendix G). **In particular, whereas TopK improves upon Gated at some sites and layers but is beaten by Gated at other sites and layers (looking at Figures 2, 14 and 15 together), JumpReLU consistently provides the best reconstruction fidelity across all sites and layers we have tested**. This consistency is valuable: it means practitioners can confidently choose JumpReLU SAEs knowing they will get state-of-the-art or better performance without needing to experiment with different architectures for different use cases. Moreover, we show that this reliable performance advantage does not come with downsides – JumpReLU SAEs are comparably interpretable to TopK and Gated SAEs, faster to train, have similar or fewer high frequency features than TopK, and perform better on our disentanglement evaluation.
>
> _Regarding the intuition behind JumpReLU versus TopK_: JumpReLU addresses a fundamental limitation of TopK that is explicitly noted in their own paper. TopK forces exactly k features to be active at every token, which is unnatural – we would expect the number of relevant semantic/causal features to vary across tokens. JumpReLU, by using thresholds rather than counts, allows this natural variation while maintaining average sparsity through the L0 penalty. This more flexible approach to sparsity likely contributes to JumpReLU's consistent performance advantages.
>
> _Regarding ablation studies_: Thank you for this valuable suggestion. We have now added ablations in Appendix H.2 (referenced from Section 5.1) that isolate our two main contributions: (a) JumpReLU activation with traditional L1 sparsity penalty, and (b) ReLU activation with our L0 sparsity penalty. Neither configuration matches the performance of the complete JumpReLU SAE with L0 penalty, demonstrating that both components are essential for the observed improvements. Regarding the suggested TopK \+ L0 ablation – we respectfully point out that this combination doesn't make sense conceptually since TopK already enforces sparsity directly through its k parameter. Adding a L0 penalty (or any other sparsity penalty) to TopK would be redundant.
>
> _On computational efficiency_: While the complexity difference between O(k log n) and O(1) might seem modest, it becomes significant in practice, particularly when training large SAEs or conducting extensive hyperparameter searches. In our setup, TopK (with AuxK) is approximately 50% slower to train at K=70 than JumpReLU, even using an approximate top-K algorithm. Moreover, TopK's computational cost scales linearly with K, while JumpReLU remains constant; this is important to keep in mind when there is still no consensus on the optimal level of sparsity required for SAEs to be useful for downstream applications.
>
> _Regarding the manual interpretability study_: We understand your concerns about potential bias in our interpretability ratings. However, we implemented rigorous controls to ensure objectivity: the study was conducted double-blind, with features randomly assigned across architectures and raters unaware of which features came from which SAE architecture during assessment. This methodology was specifically designed to eliminate the kind of bias you have identified as a concern.
>
> In conclusion, we believe our work makes substantial contributions to the field of mechanistic interpretability through SAE training. Unlike TopK or Gated SAEs, the JumpReLU SAEs introduced in this paper achieve consistently superior reconstruction fidelity across models, sites and layers, doing so through a more flexible and computationally efficient sparsity mechanism. Furthermore, our theoretical insights connecting straight-through estimators to kernel density estimation open new possibilities for training models with discontinuous loss functions, with potential applications beyond SAEs.

---

> > ### Comment · Reviewer_EMQL · 2024-11-22
> >
> > I thank the authors for the rebuttal. Most of my concerns are addressed. I have a few more comments.
> >
> > > TopK forces exactly k features to be active at every token, which is unnatural – we would expect the number of relevant semantic/causal features to vary across tokens.
> >
> > If this is your motivation, then you need to demonstrate how your method allows different numbers of relevant features to be extracted. On the other hand, I am not sure if I fully agree with you that using k features for a fixed k is unnatural, because a fixed number of features is so widely used in machine learning, such as feature selection or PCA. Using a variable k might lead to improvement, but it is still hard for me to see this from your experiment results.
> >
> > > The complexity difference between O(k log n) and O(1)
> >
> > I don't think your method is O(1), because you have an extra component that handles L0 regularization. I'd suggest compare the run time of your method and TopK, with the same fixed number of epochs and other settings.
> >
> > Like I said, I am not an expert in this field. I've read other reviewers' comments and they also raised some valid concerns. I will discuss with my fellow reviewers during the discussion period.

---

> > > ### Author Response · Authors · 2024-11-25
> > >
> > > Thank you for reading our response and for your additional comments.
> > >
> > > **Regarding computational complexity**
> > >
> > > You are correct – JumpReLU has O(n) complexity, not O(1) as we incorrectly stated. However, there remain two important practical advantages:
> > >   1. JumpReLU's complexity is independent of the desired sparsity level, unlike TopK's O(n log k)
> > >   2. Being an elementwise operation, JumpReLU is highly parallelizable.
> > >
> > > Moreover, training TopK SAEs requires the AuxK loss term which adds another O(n log k_aux) operation per step, whereas the L0 sparsity penalty used to train JumpReLU SAEs only adds another O(n), highly parallelizable, operation. In our benchmarks, this translates to TopK (with AuxK) being approximately 50% slower to train than JumpReLU at k=70.
> > >
> > > **Regarding variable feature counts**
> > >
> > > While fixed feature counts are effective in many ML applications like PCA, our goal with SAEs is to identify interpretable, causally relevant features in language model activations. Since, in natural language, tokens naturally vary in their semantic content, we expect the number of meaningful features to vary accordingly. JumpReLU allows this variation through its threshold mechanism – activations above the threshold remain active regardless of count. In contrast, TopK forces exactly k features to be active per token, potentially including weak activations or excluding strong ones to maintain this fixed count.
> > >
> > > While we could demonstrate this difference empirically by showing the distribution of active feature counts (which would show zero variance for TopK by construction), such an analysis would not tell us whether JumpReLU's variable feature counts better capture meaningful semantic variation. Instead, we believe the most compelling evidence for JumpReLU's effectiveness comes from the empirical results in Section 5, where it consistently achieves better reconstruction fidelity than both TopK and Gated SAEs across all tested layers, sites and models (Figures 2, 8, 14, 15), while maintaining comparable interpretability and improved computational efficiency.

---

> > > > ### Comment · Reviewer_EMQL · 2024-11-26
> > > >
> > > > Thank you for the response. I have no more questions at this point. I will discuss with my fellow reviewers and the AC.

---

### Official Review · Reviewer_VEDS · 2024-10-30

**Soundness:** 4
**Presentation:** 4
**Contribution:** 3
**Rating:** 8
**Confidence:** 3

**Summary:**

This paper tackles the important issue of LLM interpretability and does so by improving Sparse Autoencoders (SAEs), an unsupervised learning technique that decomposes model activations into interpretable features. This paper introduces a new activation function: jumprelu.

Jumprelu removes values below a certain greater than 0 threshold, and the threshold is trained to improve L0 (sparsity) and reconstruction fidelity.

The results are comparable to or better than leading SAE activation functions such as top-k and gated SAEs, but the computational cost is lower than top-k’s.


My **recommendation**: Accept, 8/10 (good paper).



### Why not lower score?

- The results are state of the art, and represent a pareto improvement.
- The approach is well placed in the literature, and well motivated.

### Why not higher score?

- The improvement is incremental, but not beyond comparison with alternatives.
- The application of Jumprelu to SAEs is new, but both tools were already known.

**Strengths:**

## State of the art results

Jumprelu SAEs are easier to train than top-k SAEs, as they don’t require a sorting operation to get the top-k activations at each pass.  The ***quality*** of the method is such that it attains state of the art results, and it is a ***significant*** improvement over the pareto frontier. It represents an ***original*** application of Jumprelu to LLM activations.

## Direct L0 training

Jumprelu SAEs can be trained directly on the L0 loss, which is an improvement over approximations such as the L1. L0 does not lead to activation shrinkage as is the case with L1. This approach is ***clearly*** justified and explained with the usage of a STE that allows the authors to compute pseudo derivatives.

## Evaluations: automated interpretability

The results of the automated interpretability experiments are encouraging, and they suggest a big majority of features may be automatically interpretable. This is important as the number of features represented in large language models might scale superlinearly with model size.

## Hyperparameter transfer

The authors introduce a new hyperparameter: epsilon. They use it in the definition of their pseudo derivative and note that it transfers well across model sizes. This can make scaling the method to larger models easier.


## Reproducibility

The results of this paper refer to the Gemma suite of LLMs, which is open source and allows for the reproduction of the author’s results.

**Weaknesses:**

Most of my concerns are acknowledged in the limitations section of the paper, but the following are not fully addressed.

## Downstream evaluations

Following work on sparse feature circuits (Marks et al., 2024) and using known features the authors could have included evaluations on downstream tasks, such as circuit reconstruction (faithfulness, completeness, etc..) or on the model’s ability to retrieve known features. Downstream evaluations would have strengthened the paper.

## Hyperparameter introduction

Introducing the epsilon hyperparameter without a reliable selection method, while mitigated by the empirical observation that epsilon transfers, still makes for a more complex method. Furthermore, no alternative formulations for the pseudo derivative are proposed/considered.

## No mitigations for high frequency features

While this is addressed in the limitations and future directions, there is no attempt to mitigate the presence of uninterpretable high frequency features, which represents a drawback of the proposed method. Moreover, there is no explanation for why there are more high frequency features in jumprelu SAEs than in top-k SAEs.

**Questions:**

Have you tried alternative formulations for the pseudo derivative that don’t require a hyperparameter?


Have you tried resampling (resetting) high frequency features?
This is done for dead features and can help.

Have you considered downstream task evaluation (faithfulness, completeness, etc..) of known circuits (such as IOI) using mechanistic interpretability (Marks et al., 2024)?

---

> ### Author Response · Authors · 2024-11-21
>
> Thank you for your thorough and insightful review of our paper. We particularly appreciate your recognition of our paper's technical contributions, including both the state-of-the-art performance and the principled approach to training with discontinuous loss functions.
>
> We acknowledge and appreciate the constructive criticisms you have raised. Below we address each of your main points and questions:
>
> * *Downstream evaluations.* While we would certainly have liked to include more evaluations, like the two that you suggested, our preliminary experiments suggested that these two specific evaluations lack the resolution needed to meaningfully distinguish between SAE architectures (although they can be useful, for example, for distinguishing between SAEs of differing widths within a single architecture, or e.g. for showing that SAEs of any type provide better decompositions than PCA or the neuron basis). We consider it to still be an open problem how to evaluate SAEs on downstream tasks in a way that provides sufficient resolution to distinguish between architectures, a problem which we wish to tackle in future work.
> * *Hyperparameter introduction.* While we agree that having a principled way to select ε would be valuable, we note that this parameter has a clear interpretation as a kernel bandwidth parameter (or, in the case of a rectangular kernel, also as a finite difference step-size parameter), making it more theoretically grounded than, for example, the auxiliary loss terms required by Gated SAEs, and hence easier to reason about when performing hyperparameter selection. Moreover, we strongly suspect it should be possible to develop automatic bandwidth selection methods by building on the extensive literature on bandwidth selection for kernel density estimation. (With reference to your question about alternative formulations of the pseudo-derivative, we cannot think of any formulation that doesn’t involve some sort of bandwidth parameter that controls the bias-variance trade-off that is inevitably required for estimating gradients in expectation, but are open to suggestions.) However, we did not prioritize further investigation along these lines after finding that, after applying dataset normalization, our specific choice of hyperparameters seems to transfer well across different models, sites and layers (and moreover, training is relatively insensitive to these hyperparameters). Nevertheless, we do think this would be an interesting line of further work to improve the JumpReLU training methodology.
> * *High frequency features*. We respectfully point out that we in fact find that JumpReLU SAEs typically have slightly *fewer* high frequency features than TopK SAEs (Figure 4), although we agree with the general point that the presence of such high frequency features (in both JumpReLU and TopK SAEs) is undesirable. Regarding your suggestion about resampling high-frequency features – while this is an interesting direction, we suspect that unlike dead features, high-frequency features often play an important role in reconstruction fidelity, making their resampling during training potentially disruptive. However, we believe there may be more promising approaches to addressing this issue through modifications to the sparsity penalty. Indeed, this highlights another advantage of JumpReLU SAEs over TopK SAEs – our approach allows us to explore different potentially discontinuous sparsity penalties beyond L0 (cf. Appendix F) to incorporate additional desirable properties. We are actively investigating this direction in ongoing work.

---

> > ### Comment · Reviewer_VEDS · 2024-11-25
> >
> > Dear authors,
> >
> > Thanks for addressing my points. On your response:
> > - evals: it’s useful to know that IOI type downstream tasks are not granular enough to distinguish between architectures, and it might be interesting to include it.
> > - Hyperparameter: while I agree that the bandwidth has to be set, and I am happy to hear that it is quite robust to variations, I think it could interesting to see if it can be automatically set. Is there any relationship between the norm of the weights/activation and an effective bandwidth? If the hyperparameter resists variations a rough estimate might suffice.
> > - high frequency features: it is not entirely evident that topk has more of them just by visual inspection, but it seems possible.
> >
> > With regards to my score and the other reviews, I believe this paper should be a clear accept, and I was surprised to see other reviewers disagree.
> > I would be willing to e-meet with my fellow reviewers and the area chair to make a case for this paper. I think the lack of granular enough sae evals (not just in this paper, but in the literature as a whole) undersells the contribution this paper offers.

---

> > > ### Author Response · Authors · 2024-11-26
> > >
> > > Thank you for your reply and your endorsement of the contribution made in our paper.
> > >
> > > To respond to your follow-up questions / comments:
> > >
> > > * _On the granularity of SAE evals._ To some extent this issue is already visible in the IOI evaluation results of [Makelov (2024)](https://openreview.net/forum?id=JdrVuEQih5). Although the results there do distinguish between ReLU SAEs and Gated / TopK SAEs, the distinction between Gated and TopK SAEs is less clear. The two metrics along which Gated and TopK SAEs do somewhat differ (edit magnitude, plotted vertically on Figure 1 of that paper, and mean-max cosine similarity of the Pos attribute in Figure 3), are geometric in nature and not directly related to any downstream consequences. Our disentanglement eval, shown in Figure 5(b), is an attempt to improve upon the eval in Makelov (2024), firstly by making a smaller intervention on the model, and secondly by directly measuring collateral damage inflicted by the intervention (through edit localization), instead of a geometric proxy (such as edit magnitude, used in Makelov, 2024). While we do observe meaningful distinctions between all three architectures in this evaluation, it still has limitations as detailed in Appendix J.2.2. As noted there, we conclude that the field needs substantially better methods for evaluating SAEs – methods that are both sufficiently granular and relevant to downstream applications. This has become a major focus of our ongoing research.
> > > * _Natural scales for ε._ After performing dataset normalization as described in the paper, inputs to the SAE have an average L2 norm of 1\. The SAE's task is to decompose this activation vector into a sum of component vectors, where the number of components is determined by L0 and each component has L2 norm equal to the corresponding SAE feature activation (due to our constraint enforcing unit decoder direction norms). This relationship, combined with the number of latents, sets a natural scale for typical SAE feature activations. For example, assuming roughly orthogonal components, we would expect that a L0=100 SAE would have a typical feature activation of around 0.1. (In practice, this agrees reasonably well with the typical feature activation distributions we observe in our trained SAEs.) This context provides an intuitive interpretation of our choice to set ε=0.001: it corresponds to performing KDE with a bandwidth of 0.001 to estimate probability densities that typically vary on the scale of about 0.1. While this alone doesn't definitively prove that 0.001 is the optimal bandwidth – which also depends on the smoothness of the probability distribution – it helps explain why this value is reasonable and why we should expect it to transfer well across layers, sites and models, provided we normalize the data similarly and train SAEs of similar sparsity.
> > > * _High frequency features for TopK versus JumpReLU._ In our previous response, we were alluding to Figure 4, where we plot the proportions of high frequency features for the different SAE architectures as a function of loss. We find that TopK (purple) does typically have slightly more high frequency features than JumpReLU (green), although the headline result is that both these SAE types have high frequency features at all while Gated SAEs have hardly any.
> > >
> > > We appreciate your willingness to discuss the paper's contributions with the other reviewers and area chair, and hope these clarifications are helpful in supporting that discussion.

---

### Official Review · Reviewer_NvTf · 2024-10-31

**Soundness:** 3
**Presentation:** 3
**Contribution:** 2
**Rating:** 3
**Confidence:** 3

**Summary:**

JumpReLu Sparse Autoencoders (SAE) are here introduced. Instead of using ReLU, as in the original formulation of SAE, JumpReLU can result in more sparse but strong activations. This framework is further tuned for performance, with a couple ad-hoc design choices. 1) A loss function consisting of a weighted sum of reconstruction error (L2) and sparsity (L0) is used, which attempts to maximize fidelity with as little activations as possible. 2) In order to train the threshold parameter, authors use STE. This algorithmic trick allows to estimate the gradient of the *expected* loss and thus to train JumpReLU SAE with gradient methods, even when the loss function would not provide useful gradients.

In one experiment, they show matching or slightly better results than benchmark in reconstruction fidelity for three selected layers of Gemma 2.

The authors characterized some intrinsic properties of JumpReLU SAE operations, like the proportion of feature that are very frequently active and interpretability of results. They also performed an editing experiment. In all these cases, results are similar to previous work.

**Strengths:**

The paper has a clear step-by-step presentation of SAE and of the algorithmic features utilized, moving from general to specific. By introducing general equations, including the loss function and SAE architectures, it provides a solid basis for understanding. Each methodological detail is thoroughly explained, making the thought process reproducible and leaving little open to interpretation. The authors effectively clarify the SAEs used, and discuss the broader implications of results, adding depth and impact to their findings. I also enjoyed the slightly informal writing style, it reads very clear.

The submission is also technically correct, with the exception of some points (see below).

**Weaknesses:**

My assessment is that unfortunately this paper offers a limited incremental contributions, as it builds on ReLU SAE with the introduction of JumpReLU. It does add two interesting algorithmic tweaks (a custom loss function and STE for gradients) but they are not central. These adjustments fall short of significantly advancing the state of the art within the SAE domain. The authors’ claim of state-of-the-art performance is based on a single experiment with one model, where selected layers appear cherry-picked, with only marginal improvement. Other experiments on encoding explainability fail to surpass benchmarks.

Changing my opinion on the testing being too selective would imply seeing results across the layer stack (how did you select layer 9-20-31?) and on a second model other than Gemma 2 9B. I believe it's too big of an ask for revision, so I am a bit pessimistic about it.
Additionally, figure legends and labels are frequently inadequate, contrasting with the otherwise high quality of the text.

**Questions:**

- Figure 1: What do the blue and red colors represent? Please clarify the legend. also please label the axes.
- Why is JumpReLU restricted to a positive threshold?
- Figure 2: This figure is poorly explained. To what does "Delta LM loss" refer here? This term is introduced too early in the paper (Figure 2 is on page 2, while Delta LM loss is introduced on page 6), making the figure difficult to interpret as presented. Please state that axes are log scale.
- The preliminary definition of SAE is incorrect. The encoder should map to a smaller latent space, m<<n . Explicitly stating the activation function in Equation 1 would also be helpful.
- Why were layers 9, 20, and 31 specifically selected?
- Figure 3 please label axes.
- Figure 4: indicate x is log scale, while y is not.
- An example figure of the rating procedure would be helpful, even as an appendix item.
- The results on automated interpretability are well explained but would be clearer if presented in a table, as is standard in this type of publications.
- Typo in Line 185 "be actually be".
- Line 216: refer to the relevant section.
- Page 27-28: Algorithms should be presented as pseudocode, not as actual implementations, as this style is harder to understand in a CS paper. Implementation details can be saved for a code-publishing section.

---

> ### Author Response · Authors · 2024-11-21
>
> Thank you for your thorough review of our paper, we are heartened by your positive comments on the paper’s clarity and technical correctness. We have carefully considered the weaknesses you have listed and believe we can effectively address both your primary concerns: (a) that you feel our testing has been too selective, and (b) that the contribution of this paper is too incremental.
>
> On (a), the selectiveness of our testing, we believe this is straightforward to address. Firstly, we do indeed compare JumpReLU against TopK and Gated SAEs on another model, Pythia 2.8B, in Appendix G (and signposted in the Conclusion section) precisely to address the concern that our results may not generalize beyond Gemma 2\. Secondly, for each tested layer, we conduct evaluations across three different activation sites (residual stream, MLP output, and attention output), showing the JumpReLU SAEs consistently provide the most faithful reconstructions across multiple sites, i.e. not just for residual stream activations. Finally, our choice of layers 9, 20 and 31 of Gemma 2 9B was made because these are evenly spaced layers roughly 1/4, 1/2 and 3/4 through this 42-layer model, i.e. to check that our results generalise to different layers in the model. We have added text at the start of Section 5 to clarify why we chose these layers in the paper. Most importantly, we emphasize that we chose these to evaluate the SAEs on these layers _prior_ to running the experiments, i.e. we did not cherry-pick these layers to improve our results.
>
> Regarding (b) your concern about incremental contribution, we believe our work represents a meaningful advance in both theoretical and practical aspects of the field. Theoretically, we introduce a principled approach to training through discontinuous loss functions using straight-through estimators, which has broader implications beyond just SAEs. Empirically, our extensive evaluations demonstrate that JumpReLU SAEs deliver superior or equivalent reconstruction fidelity compared to existing approaches across all tested scenarios – spanning different layers, activation sites, and model architectures. Although there may be some sites and layers where the difference between JumpReLU and TopK SAEs isn’t massive, the key strength of our approach lies in its consistency and reliability: as Figures 2, 8, 14 and 15 show, **JumpReLU SAEs reliably provide the best reconstruction fidelity across models, sites and layers. This cannot be said of TopK and Gated SAEs, which trade places between different sites and layers.** This reliability of JumpReLU SAE performance is particularly valuable from a practical standpoint: researchers can adopt JumpReLU SAEs with confidence, knowing they will achieve at least state-of-the-art performance without the need to experiment with different architectures for different applications. Furthermore, these improvements come with significant practical benefits – JumpReLU SAEs are more efficient to train than alternatives (avoiding both the auxiliary terms needed for Gated SAEs and expensive TopK operations), demonstrate comparable or better interpretability metrics, and show stronger performance on disentanglement tasks. We believe this combination of theoretical insight, reliable performance improvements, and practical advantages represent a meaningful contribution to the problem of training faithful SAEs to decompose LM activations, a problem that has received much attention this year in the field of mechanistic interpretability.
>
> Addressing your specific questions and suggestions:
>
> - _Improvements to figures_ Thank you for your feedback, We have updated all figures accordingly and added an example of the dashboards used in the interpretability study (Figure 12).
>
> - _Why is JumpReLU restricted to a positive threshold?_ While negative thresholds are possible, we follow the convention from Bricken et al. (2023) of looking for non-negative linear combinations of features, stemming from the superposition hypothesis and empirical results suggesting LMs could use negative feature directions to represent unrelated features. Exploring negative thresholds could nonetheless be interesting future work.
>
> - _The preliminary definition of SAE is incorrect:_ We respectfully disagree. While traditional autoencoders often reduce dimensionality, all work on LM SAEs since their introduction has used an overcomplete basis (m\>\>n) to discover interpretable features. For example, our experiments use m=131,072 features for n=3,584 dimensional activations.
>
> - _Presentation of algorithms as pseudocode:_ While we appreciate algorithms are typically better in CS-style pseudocode, in this case we believe the implementation details are more valuable. The mathematical formulation is already specified in Equations 8-12. The code specifically demonstrates how to implement custom gradients in an autograd framework – a crucial detail that would be obscured by traditional pseudocode notation.

---

> > ### Comment · Reviewer_NvTf · 2024-11-24
> >
> > Thank you for the revised manuscript and the multiple clarifications. While, as detailed in my assessment below, several concerns remain for me, I appreciate the improvements made and recognize the effort invested in this revision.
> >
> > Regarding your choice of Gemma 2 with 9B layers, I’d like to clarify my earlier comment about the layers appearing “cherry-picked.” The use of this term was unfair. I do not question the integrity of your selection process and trust that it was guided by a structured plan rather than performance-driven cherry-picking. My concern, instead, is that the analysis plan appears underdeveloped due to sparse testing.
> > Another example, Fig. 5(b) uses Layer 9 for evaluation, supported by an explanation in J.2.1 that describes this choice as an informed guess. However, without broader comparisons, we cannot ascertain whether Layer 9 is a particularly high- or low-performing choice overall. This lack of context also limits our understanding of how SAE types would compare under alternative choices, such as an actual "optimal" layer for that task.
> >
> > On the issue of incremental contribution, I acknowledge and appreciate your defense of the theoretical and practical benefits of your approach. However, this point largely relies on subjective interpretation from diverse readership. To this reviewer, applying straight-through estimators and a discontinuous LF is a combination of methods and a good engineering choice in design when several alternative options are offered, albeit of limited theoretical gain unless the manuscript is built around theoretical analysis of this synergy, which is not.
> >
> > Empirically, I agree that that JumpReLU SAEs deliver superior or equivalent reconstruction fidelity compared to existing approaches. However, your results show two scenarios where, under the Delta LM Loss metric, JumpReLU performs slightly better than TopK in layer 9 (Gemma) and layer 15 (Pythia, where no other layer is tested). At the same time, the Editing Success Rate metric for Gemma seems to favor TopK, as its distance to "perfect" in the ROC is smaller in norm. Taken together, while there is evidence of improvement, the effect size is relatively small and use-dependent.

---

> > > ### Author Response · Authors · 2024-11-26
> > >
> > > Thank you for considering our reply and your additional comments. We appreciate your clarification regarding layer selection and acknowledge the inherently subjective nature of assessing a paper's contribution.
> > >
> > > Our evaluation strategy – covering 3 layers, 3 activation sites, and a second model – reflects a balance between demonstrating generality and maintaining clarity in presentation. Furthermore, training and evaluating SAEs at this scale is already computationally intensive, and we note that our current coverage already exceeds that of comparable prior work. For instance, Gao et al (2024) evaluate TopK SAEs against alternatives on a single layer and site across two models, whereas we compare architectures across early, middle, and late layers at all three principal sites where prior work has trained SAEs.
> > >
> > > We would appreciate if you could clarify your observation about JumpReLU performing "slightly better than TopK in layer 9 (Gemma) and layer 15 (Pythia)" so that we can respond more precisely. For Gemma layer 9, we present results across three different sites (residual stream, MLP output, and attention output) in Figures 2, 14, and 15, and it would be helpful to understand which comparison you're referring to. Additionally, the curves appear to show JumpReLU consistently providing the best delta LM loss across different L0 values, so it is unclear to us which are the two scenarios you are alluding to.

---

> > > > ### Comment · Reviewer_NvTf · 2024-11-28
> > > >
> > > > Thank you.
> > > > I am not sure there is a discrepancy in interpretation regarding Figure 2. I acknowledged that Jump ReLU SAEs performs slightly better than TopK for SAEs trained on the residual stream after layers 9. That is aligned with your interpretation of the results that I can read in the figure caption: "reconstruction fidelity [...] equals or exceeds Gated and TopK SAE". I am not alluding to other scenarios here.
> > > >
> > > > At the same time, the Editing Success Rate (Figure 5b) seems to favor TopK, whose best point has smaller distance to "perfect" in the ROC (distance vector norm). It looks farther because the x axis is stretched.
> > > >
> > > > Taken together, I wrote that the overall improvement of the proposed novelty seems limited. I will discuss with my fellow reviewers at the next phase.

---

### Official Review · Reviewer_xaZF · 2024-10-31

**Soundness:** 1
**Presentation:** 2
**Contribution:** 1
**Rating:** 3
**Confidence:** 4

**Summary:**

For LM activation decomposition, the paper proposes to use JumpReLU activation and $L^0$ norm instead of $L^1$ norm for sparse autoencoders. The proposed method obtains similar results as the baseline models.

**Strengths:**

* Good discussion about the limitations and the future works.

**Weaknesses:**

* The motivation of replacing ReLU by JumReLU activation for SAEs is not lear, the reason is not even explained in the paper.
* The idea of using JumReLU activation for SAEs is already presented in Gated SAEs, if I am understand it well.
* The paper is not well-written and the notations are not rigorous. To name a few, in Equation 1, $\mathbf{f}$ is used both for the function and the vector. For the notations of norm, please use $L^p$ instead of $Lp$.
* While the paper proposes to change the activation function, it proposes also to change the regularization norm type. However, there are no ablation study has been conducted in the experiments, so we have no clue from which part of the contributions comes from the improvement.
* The performance improvement of the method is not evident.

I strongly recommend the authors rework the paper.

**Questions:**

* Can the authors explain the motivations behind the JumReLU activation?
* Can the authors conduct the ablation study as stated in the weakness?
* Can the author give more context about the problem to solve, for example, why we need the sparse autoenocder to evaluate LM?

---

> ### Author Response · Authors · 2024-11-21
>
> Thank you for your detailed feedback, which has helped us identify important areas for improvement. We have made concrete changes to address your concerns about motivation, ablation studies, and technical presentation, as detailed below.
>
> Addressing your questions first:
> * *What are the motivations behind the JumpReLU activation?* The motivation for replacing ReLU with JumpReLU is illustrated in Figure 1, which shows why this change is necessary: ReLU cannot effectively separate active from inactive features without introducing false positives or underestimating activation magnitude (shrinkage). When a feature's pre-activations are high where the feature is present but low (but not always negative) where the feature is absent, ReLU either allows false positives or requires bias adjustments that systematically underestimate feature magnitudes. JumpReLU solves this by providing a threshold that screens out false positives without affecting magnitude estimation. We have added a sentence to the introduction, signposting that the motivation behind the JumpReLU activation is explained in Figure 1\.
> * *Can the authors conduct the ablation study as stated in the weakness?* We have added an ablation study in Appendix H.2 (referenced from Section 5.2 in the main text) that separates our two main changes. Specifically, we evaluate:
>   1. Using JumpReLU activation while keeping the L1 sparsity penalty from prior work
>   2. Keeping the ReLU activation while introducing our L0 sparsity penalty
>
>   Neither ablation matches the performance of JumpReLU SAEs trained with a L0 penalty, demonstrating both components are necessary for the improvements shown.
>
> * *Can the author give more context about the problem to solve, for example, why we need the sparse autoencoder to evaluate LM?* Our Introduction and Related Work sections provide extensive context for SAEs in LM interpretability. Specifically, we explain that SAEs help identify interpretable directions in LM activations, which are valuable for tasks like circuit analysis and model steering. We also discuss how many concepts appear to be linearly represented in LM activations, making dictionary learning approaches like SAEs particularly promising for understanding and controlling these models. These sections cite the key foundational works that established the value of SAEs for LM interpretability. Since our paper's contribution is a methodological improvement to SAE training rather than introducing SAEs for LMs, we believe this level of context appropriately situates our work while allowing us to focus on our technical contributions.
>
> Additionally, to address the weaknesses you have listed (not already covered above):
>
> * *Notation*. Regarding notation, we acknowledge that $L^p$ is standard mathematical notation. However, our use of L0 and L1 follows established convention in the language model SAE literature (e.g. Bricken et al, 2023; Templeton et al, 2024) where this notation is widely used and understood by the community. For Equation 1, we use $\\mathbf{f}$ to denote both the function and its output for clarity of exposition and to maintain consistency with prior work.
> * *The idea of using JumpReLU activation for SAEs is already presented in Gated SAEs*. While it is true that Gated SAEs can be shown to implement a JumpReLU activation function (as noted in the Gated SAEs paper), our key contribution is not the activation function itself but rather introducing a principled method for training SAEs with discontinuous activation functions. Instead of relying on auxiliary loss terms to provide indirect gradients for training the threshold (as Gated SAEs do), we derive and directly minimize the expected loss of the SAE, despite the discontinuous nature of the JumpReLU function. As our evaluation shows, this more principled training approach leads to consistently better reconstruction fidelity than Gated SAEs at any given level of sparsity.
> * *The performance improvement of the method is not evident.* Our evaluations show that JumpReLU SAEs consistently match or exceed the reconstruction fidelity of both Gated and TopK SAEs across all tested combinations of layers, sites and models (including Pythia-2.8B results in the appendix). While TopK sometimes beats Gated and sometimes loses to Gated (considering all sites and layers shown in Figures 2, 14 and 15), JumpReLU reliably provides the best sparsity-fidelity trade-off regardless of where in the network it is applied. This consistency is valuable: it means practitioners can confidently choose JumpReLU SAEs knowing they will get state-of-the-art or better performance without needing to experiment with different architectures for different use cases. Moreover, we show that this reliable performance advantage does not come with downsides – JumpReLU SAEs are comparably interpretable to TopK and Gated SAEs, faster to train, have similar or fewer high frequency features, and perform better on our disentanglement evaluation.

---

> ### Author Response · Authors · 2024-11-25
>
> Before this phase of the rebuttal period ends, we wanted to ask the reviewer whether we have addressed your concerns with our work?

---

### Meta-Review · Area_Chair_cuuf · 2024-12-17

**Metareview:**

This work introduces JumpReLU SAEs, which the authors claim achieve state-of-the-art reconstruction fidelity at a given sparsity level on Gemma 2 9B activations, surpassing recent advances such as Gated and TopK SAEs. The authors also claim that these gains do not compromise interpretability, as confirmed by both manual and automated interpretability analyses. Specifically, JumpReLU SAEs are derived from standard (ReLU) SAEs by replacing the ReLU with a discontinuous JumpReLU activation function, while maintaining similar training and runtime efficiency. By employing straight-through estimators (STEs), this approach aim to demonstrate that training JumpReLU SAEs effectively is possible despite the discontinuity introduced by the JumpReLU function during the forward pass. Furthermore, STEs are utilized to train L0 directly for sparsity instead of relying on proxies such as L1, thereby circumventing issues like shrinkage.

According to the reviewers the pros and cons were
Pros:
+ Novel Activation Function: The paper introduces the JumpReLU activation function, a variation of ReLU, designed to improve sparse autoencoder (SAE) reconstruction fidelity while maintaining interpretability.
+ Direct L0 Sparsity Training: The authors propose a principled use of straight-through estimators (STEs) to directly optimize L0 sparsity, avoiding issues like shrinkage associated with L1 proxies.
+ Empirical Comparisons: The paper evaluates JumpReLU SAEs against baseline methods such as Gated and TopK SAEs, showing comparable or slightly better reconstruction fidelity in specific scenarios.

Cons:
- Incremental Contribution: Reviewers widely agree that the paper’s contribution is incremental, primarily building on existing SAEs (e.g., TopK, Gated SAEs) and introducing minor modifications. The use of JumpReLU and STEs is considered more of an engineering tweak than a significant theoretical advance.

- Limited Empirical Significance: The reported improvements in reconstruction fidelity are marginal, with some metrics (e.g., Editing Success Rate) favoring TopK SAEs. The authors' experiments focus on limited layers (e.g., layers 9, 20, 31) and models (Gemma 2 9B), leading to concerns about cherry-picking results and generalizability.

- Incomplete Ablation Studies: The absence of thorough ablation studies makes it unclear whether the improvements stem from JumpReLU, the L0 penalty, or their combination. Reviewers note the need for additional configurations, such as TopK combined with L0 regularization, for a fair comparison.

- Evaluation Limitations: Downstream evaluations (e.g., circuit reconstruction) are missing, weakening the practical impact of the method.
Automated and manual interpretability analyses, while presented, are not sufficient to distinguish JumpReLU’s effectiveness from baseline methods.

- Lack of Theoretical Depth: The paper lacks a formal theoretical justification for why JumpReLU should outperform alternatives like TopK or Gated SAEs. The proposed method’s novelty relies more on existing tools (STEs, JumpReLU activation) than new theoretical insights.

**Additional Comments On Reviewer Discussion:**

The authors response did mitigate some concerns but the discussion among reviewers reveals significant disagreements. One reviewer (VEDS) supports the paper, arguing that its contribution lies in enabling principled training of SAEs with discontinuous activation functions. However, VEDS acknowledges the lack of sufficiently granular evaluation metrics in the broader SAE literature. Reviewers EMQL and NvTf remain unconvinced, noting the incremental nature of the contribution, weak empirical gains, and insufficient ablation studies. Both reviewers highlight the small effect size of improvements relative to baselines like TopK SAEs. I encouraged reviewers to reconcile their perspectives, but the consensus leans toward rejection due to the limited impact and lack of strong empirical or theoretical contributions.

---

### Decision · Program_Chairs · 2025-01-22

Reject